# Mcam inhibits macrophage-mediated development of mammary gland through non-canonical Wnt signaling

Xing Yang [1,2,3,9], Haibo Xu [3,9], Xu Yang [3], Hui Wang [3], Li Zou[3], Qin Yang[3], Xiaopeng Qi[3], Li Li[4], Hongxia Duan [5], Xiyun Yan [5], Nai Yang Fu[6,7], Jing Tan [1,2] ✉, Zongliu Hou [1,2] ✉ & Baowei Jiao [3,8] ✉

While canonical Wnt signaling is well recognized for its crucial regulatory functions in cell fate decisions, the role of non-canonical Wnt signaling in adult stem cells remains elusive and contradictory. Here, we identified *Mcam*, a potential member of the non-canonical Wnt signaling, as an important negative regulator of mammary gland epithelial cells (MECs) by genome-scale CRISPR-Cas9 knockout (GeCKO) library screening. Loss of Mcam increases the clonogenicity and regenerative capacity of MECs, and promotes the proliferation, differentiation, and ductal morphogenesis of mammary epithelial in knockout mice. Mechanically, Mcam knockout recruits and polarizes macrophages through the Il4-Stat6 axis, thereby promoting secretion of the non-canonical Wnt ligand Wnt5a and its binding to the non-canonical Wnt signaling receptor Ryk to induce the above phenotypes. These findings reveal Mcam roles in mammary gland development by orchestrating communications between MECs and macrophages via a Wnt5a/Ryk axis, providing evidences for non-canonical Wnt signaling in mammary development.

The canonical Wnt pathway plays a critical role in various developmental processes including stem cell maintenance, self-renewal, and differentiation of the mammary gland[1-3]. Non-canonical ligands (e.g., Wnt5a, Wnt5b, and Wnt11) bind to their receptors (e.g., FZD receptors and/or Ror1/Ror2/Ryk co-receptors), thereby activating PCP/RTK or $Ca^{2+}$ signaling cascades[4] and regulating cell polarity during morphogenesis, cancer, inflammation, and neurodegenerative disease[5]. For adult stem cells, the previous reports demonstrated indirectly regulatory roles of non-canonical Wnt signaling in neural stem cells and hematopoietic stem cells, e.g., by promoting the Notch signaling pathway[6] or inhibiting the canonical Wnt signaling[7], however, the functions of non-canonical Wnt signaling in adult stem cells have not been fully established yet. Moreover, while canonical Wnt signaling has been explored in both normal and tumor stem cell microenvironments, non-canonical Wnt signaling has only been explored in regard to tumor microenvironments[4,8]. At present, there are few reports on the role of non-canonical Wnt signaling pathways in the normal adult stem cell microenvironment.

Mcam (also known as CD146) is a melanoma cell adhesion molecule, originally identified as a tumor marker for melanoma[9]. However,

[1]Yan'an Hospital Affiliated to Kunming Medical University, Kunming, Yunnan 650051, China. [2]Key Laboratory of Tumor Immunological Prevention and Treatment of Yunnan Province, Kunming, Yunnan 650051, China. [3]Key Laboratory of Genetic Evolution & Animal Models (Chinese Academy of Sciences), Chinese Academy of Sciences, Kunming, Yunnan 650201, China. [4]Research Center of Stem cells and Ageing, Chongqing Institute of Green and Intelligent Technology, Chinese Academy of Sciences, Chongqing 400714, China. [5]Institute of Biophysics, Chinese Academy of Sciences, Beijing 100000, China. [6]Cancer and Stem Cell Biology Program, Duke-NUS Medical School, Singapore 169857, Singapore. [7]ACRF Cancer Biology and Stem Cells Division, The Walter and Eliza Hall Institute of Medical Research, Parkville, VIC 3052, Australia. [8]KIZ-CUHK Joint Laboratory of Bioresources and Molecular Research in Common Diseases, Kunming Institute of Zoology, Chinese Academy of Sciences, Kunming, Yunnan 650201, China. [9]These authors contributed equally: Xing Yang, Haibo Xu. ✉e-mail: kmtjing@sina.com; HZL579@163.com; jiaobaowei@mail.kiz.ac.cn

recent evidences indicates that Mcam is not only an adhesion molecule but also actively involved in a variety of biological processes, such as development, cell migration, mesenchymal stem cell differentiation, and immune response[10], and increasingly recognized as a novel biomarker of angiogenesis and cancer[11]. Mcam is also a cellular surface receptor of multiple ligands, including certain growth factors and extracellular matrixes[12,13]. However, although Mcam plays an important role in breast cancer[14], its role in the regulation of mammary gland epithelial cells (MECs) and normal organ development of mammary glands is unclear.

An increasing number of lineage cell-fate determination factors and potential niche regulators of MECs have been revealed recently[15]. While immune cells[16,17] and fibroblasts[18,19] in the microenvironment are considered critical players in MECs, it is unclear how MECs control their interactions with these cells to maintain stem cell activity. Although several studies have explored the potential role of non-canonical Wnt in mammary stem cells (MaSCs)[20], the functions of non-canonical Wnt signaling remain controversial. For instance, TGF-β regulated the expression of Wnt5a directly in the mammary gland. While Wnt5a inhibited mammary ductal branching[21,22], Wnt5a was also demonstrated promoting roles for MaSC/progenitor proliferation[23]. Thus, the molecular mechanisms underlying the effects of Wnt5a need to be further characterized.

In this work, we find that as a non-canonical Wnt receptor, Mcam regulates the communications between macrophages and MECs, thereby maintaining MEC regeneration and subsequently promoting mammary ductal morphogenesis, providing evidence of non-canonical roles for adult stem cells.

## Results

### Mcam as a potential regulator of MECs by GeCKO screening

To identify potential regulatory factors of MECs, we performed clone formation assay[24] and transplantation assay[25,26] for screening using a pooled mouse GeCKO library containing 130 209 single guide RNAs (sgRNAs), targeting 20 611 protein-coding genes, and 1000 control sgRNAs (non-targeting sgRNAs) in the mouse genome[27]. The mammary progenitor-enriched population was isolated using standard protocols[24] and transduced with a lentivirus prepared by the GeCKO library. Multiplicity of infection (MOI) was tested and applied to ensure that each cell contained a single sgRNA. The non-transfected cells were screened out by puromycin. After 10-14 days, we selected large clones (diameter ≫ 100 μm, enriched with mammary progenitor cells) for deep sequencing. The stably transfected cells were used for a 5-week transplantation assay, and the regenerated mammary tissues were collected for deep sequencing (Fig. 1a). Thus, based on the above methods, we performed gene enrichment of representative sgRNAs from plasmids, stably transfected cells, large clones, and regenerated mammary tissues.

For sequencing, we first compared the overall distributions of sgRNAs from all samples to demonstrate the dynamics of the sgRNA library (Fig. S1a) and validate the sequencing quality. The detected number of sgRNAs in the plasmids was close to that in the original library, showing that most sgRNAs in the library were represented (Figs. S1b and S1c). The sgRNAs in the large clones and regenerated mammary tissues, retained less sgRNAs, reflecting that sgRNAs were enriched in these samples. The global patterns of sgRNA distribution in different samples were distinct, as evident by the strong shifts in respective cumulative distribution functions (Fig. S1d). In addition, we compiled a list of genes (Supplementary Dataset 1) showing enrichment of 1, 2, or 3 or more sgRNAs, along with the proportion of samples where these genes demonstrated enrichment (Fig. S1e). Based on MAGeCK[28] (http://bitbucket.org/liulab/mageck-vispr), we obtained a candidate gene list properly accounted for mammary gland morphogenesis and/or MEC clonogenicity and regenerative capacity, and found Mcam was one of the common genes in all conditions (Fig. S2a).

By further analysis, we found multiple sgRNAs targeting Mcam are significantly enriched in more than half of the regenerated mammary tissue (Fig. S2b) with two or more sgRNAs targeting Mcam enriched in each sample (Fig. S2c).

The candidates were selected according to the expression profile based on single-cell sequencing data of MECs[29], and sgRNAs ranked in each sample. The single-cell sequencing data showed that Mcam was highly expressed in basal cells (Fig. 1b). To validate this result, we collected basal (Lin⁻CD24⁺CD29⁺) and luminal (Lin⁻CD24⁺CD29⁻) cell populations using fluorescence-activated cell sorting (FACS) (Fig. S3a, b), with quantitative real-time polymerase chain reaction (qRT-PCR) confirming the high expression of Mcam in the basal cell population (Fig. 1c). Immunofluorescence (IF) staining showed that Mcam was co-localized with K14 (basal cell marker) but not with K18 (luminal cell marker) (Fig. 1d). Taken together, Mcam was identified as a potential key regulator of MECs.

### Loss of Mcam enhances mammary gland development

To further investigate the role of Mcam in regulating MEC proliferation and mammary gland development in vivo, we firstly examined Mcam expression pattern in four stages of mammary gland development, and found Mcam highly expressed in puberty (4-5 weeks), pregnancy, and lactation (Fig. 1e). Then we bred Mcam flox mice[30] with MMTV-Cre mice to generated Mcam^fl/fl/MMTV-Cre mice (cKO mice)[31] and K14-Cre mice to generate Mcam^fl/wt/K14-Cre mice (heterozygous cKO mice) with specific loss of Mcam in MECs or basal cells. Mcam was specifically depleted in MECs or basal cells based on qRT-PCR (Figs. 1f and S3c), IF staining (Figs. 1g and S3d), and western blot analysis (Figs. 1h and S3e) of whole mammary tissues. We next explored the mammary duct phenotype. Loss of Mcam significantly lengthened mammary ducts at the puberty stage (5-weeks old), and therefore increased mammary duct occupation in the fat pad (Figs. 1i, j and S3f, g). In addition to ductal length, mammary duct and terminal end bud (TEB) thickness increased significantly (Figs. 1k, l and S3h–j). The above effects were not observed in adult mice (8-weeks old) (Fig. S3k), which could be due to rare expression level of Mcam at this period (Fig. 1e).

These results showed that Mcam loss enhances mammary gland development, and this phenotype is likely due to increased proliferation and differentiation of the mammary epithelium. Based on IF staining with Ki67 (cell proliferation marker), both basal and luminal cells demonstrated a much higher percentage of Ki67-positive cells upon loss of Mcam (Figs. 2a, b and S4a–e), which is also supported by the enrichment of DNA replication pathway in cKO mice based on RNA-seq data (Fig. S4f). All cell differentiation markers, including Stat5, Elf5, Foxa1, Esr1, and Gata3[32], were increased at the mRNA level (Fig. S4g). Immunohistochemistry (IHC) staining showed PR⁻ and PR⁺ cells occurred in cKO mice by MMTV-Cre, implying Mcam loss promoting cell differentiation (Fig. S4h). The induction of both cell proliferation and differentiation suggests that loss of Mcam enhances mammary duct morphogenesis.

### Loss of Mcam promotes the regeneration of mammary gland

To investigate whether the above phenotype is driven by MaSCs, we explored changes in the MaSC-enriched cell population (Lin⁻CD24⁺CD29⁺) in cKO mice by MMTV-Cre. Compared to wild-type (WT) mice, the cKO mice exhibited a three-fold higher basal population (Figs. 2c, d and S5a, b), indicating that Mcam loss induces the basal cell compartment containing MaSCs. In addition, we examined the expression of stemness-related genes including Procr, K14, Lgr5, and sSHIP in basal cells derived from the mammary tissues of WT and cKO mice, and found these genes highly expressed in Mcam cKO mice (Fig. S5c), supporting the potential regulatory roles of Mcam for MaSCs.

To determine whether Mcam negatively regulates MaSC function in vivo, we carried out limiting-dilution transplantation assays in non-obese diabetic/severe combined immunodeficiency (NOD/SCID) mice

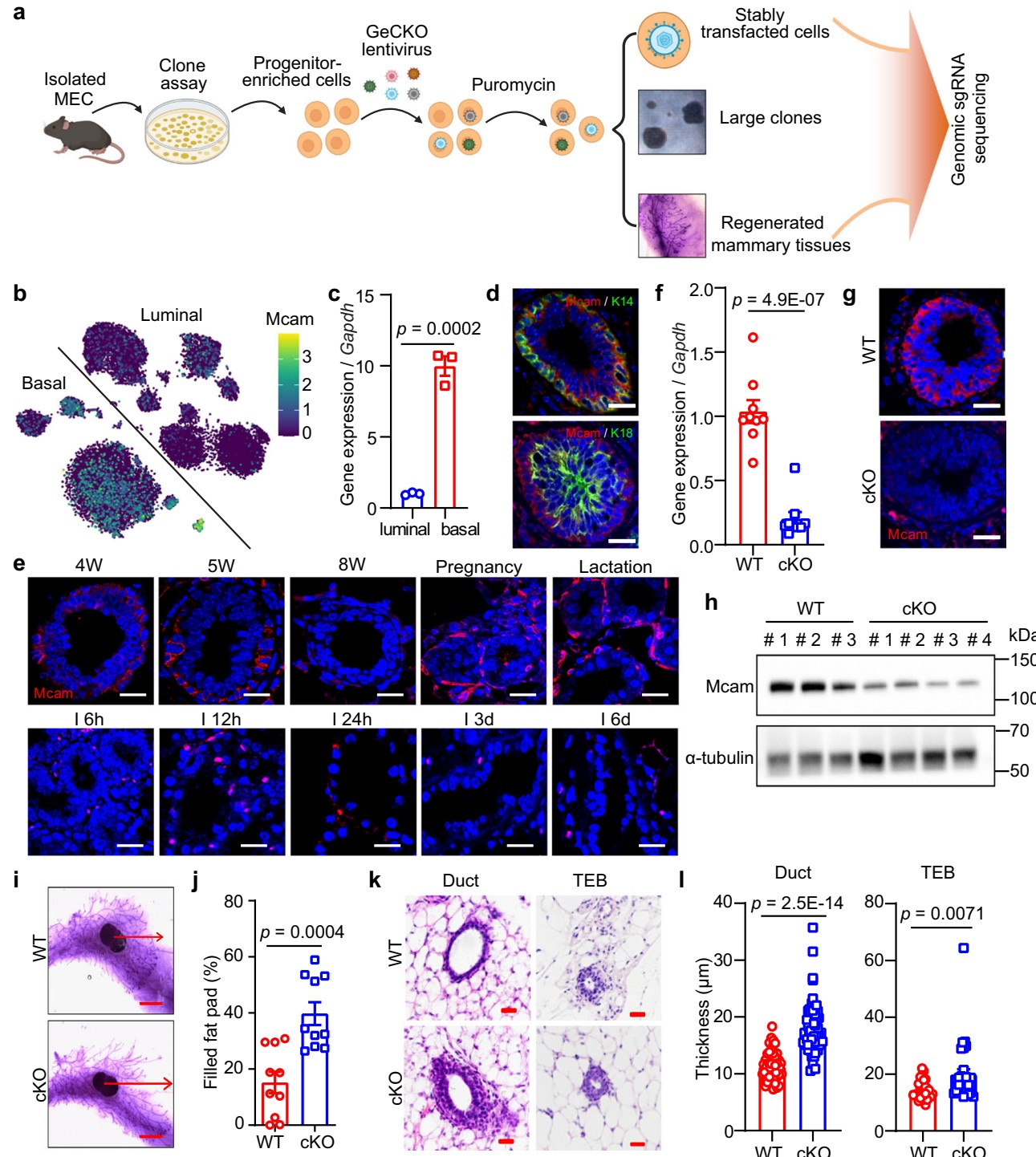

**Fig. 1 | Mcam is a negative regulator of MECs and mammary gland development. a** Schematic of screening strategy. Created with BioRender.com. **b** tSNE plot showing expression pattern of Mcam in different lineages according to single-cell sequencing data[29]. **c** qRT-PCR analysis of *Mcam* mRNA expression in different subpopulations of MECs shown in Figs. Sa-3b. n = 3 biological replicates. **d** Immunofluorescence (IF) images of Mcam co-stained with basal marker K14 and luminal marker K18. Nuclei were stained in blue by DAPI. Scale bar, 20 μm. **e** IF images of Mcam staining at various developmental stages of mammary gland. 4 W, 4-week old; I, involution. Scale bar, 20 μm. Representative images were taken from 3 mice for each stage. **f** qRT-PCR analysis of *Mcam* mRNA expression in MECs from WT and cKO mice by MMTV-Cre. n = 9 mice per genotype. **g** IF staining showing knockout efficiency of Mcam in WT and cKO mice. Scale bar, 20 μm. **h** Western blotting showing Mcam protein expression in WT and cKO mammary tissues. 3 mice for WT group and 4 mice for cKO group. **i, j** Representative images (**i**) and their quantification (**j**) of mammary duct extension of whole-mount staining in WT and cKO mice at 5 weeks of age. Arrow, length of epithelial extension. Scale bar, 5 mm. n = 10 mice in each group. **k** Representative images of mammary tissues in 5-week-old WT and cKO mice by H&E staining. Scale bar, 20 μm. **l** Quantification of mammary duct and TEB thickness in WT and cKO mice. n = 50 ducts or n = 30 TEBs in WT and cKO mice. Data are means ± SEM. Two-sided Student's *t* test was used to evaluate statistical significance. Source data are provided as a Source Data file.

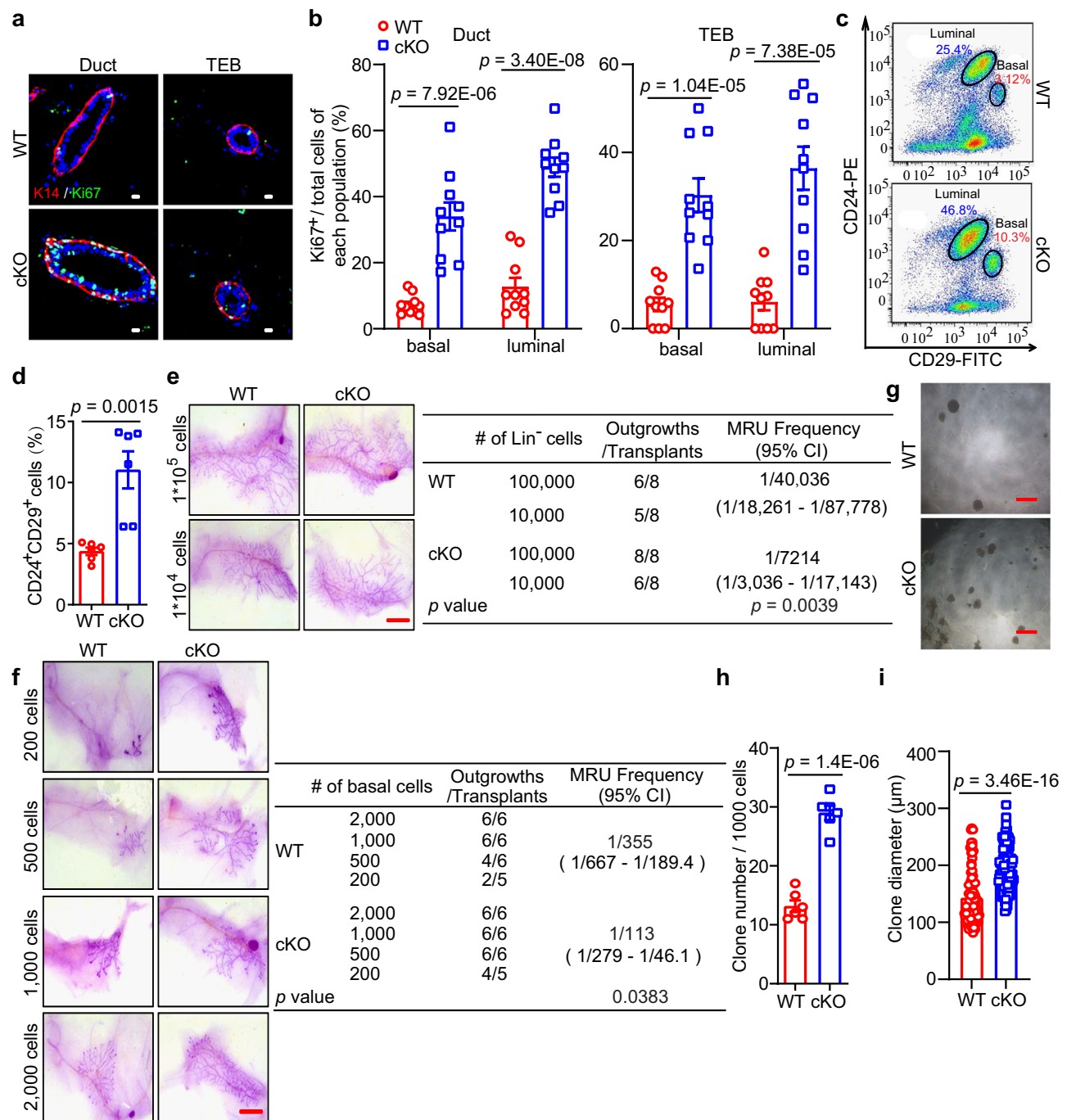

**Fig. 2 | Loss of Mcam promotes MEC regenerative capacity. a** Immunostaining of K14 (red) and Ki67 (green) expression in mammary ducts and TEBs of WT and cKO mice by MMTV-Cre. Scale bar, 10 μm. **b** Ki67+ cells in basal and luminal cell populations of mammary ducts and TEBs in WT and cKO mice. $n = 10$ sections in each group. **c, d** FACS analysis (**c**) and statistical ratios of basal cell population (Lin−CD24+CD29+) and luminal cell population (Lin−CD24+CD29−) (**d**) in WT and cKO mice. $n = 6$ mice for each genotype. **e** Representative images (left) and the reconstitution efficiency at limiting dilution (right) of whole-mount-stained mammary outgrowths derived from transplantation of isolating WT or cKO Lin− cells and harvested at 6 weeks after transplantation. Scale bar, 5 mm. $n = 8$ mice for each group. **f** Representative images (left) and the reconstitution efficiency at limiting dilution (right) of whole-mount-stained mammary outgrowths derived from transplantation of WT or cKO basal cells and harvested at 6 weeks after transplantation. Scale bar, 5 mm. $n = 5$ in 200-cell of cKO group, $n = 6$ mice in the other groups. **g–i** Representative images (**g**), clone numbers (**h**, $n = 6$ biological replicates), and clone diameters (**i**, $n = 93$ for WT and $n = 125$ for cKO by MMTV-Cre) of colonies formed by MECs derived from WT and cKO mice. Scale bar, 500 μm. Data are means ± SEM. Two-sided Student's $t$ test was used to evaluate statistical significance. Source data are provided as a Source Data file.

using isolated Lin− epithelial cells and basal population, primary MECs, respectively. All isolated Lin− epithelial cells and basal cells (regenerated mammary tissues were harvested at 6 weeks after transplantation), primary MECs (regenerated mammary tissues were harvested at 5 weeks after transplantation) from cKO mice displayed a significantly

higher rate of successful engraftment and more extensive mammary outgrowth than the WT cells (Figs. 2e, f and S5d). To illustrate whether Mcam directly regulates the MEC clonogenicity, we performed an in vitro clone assay using primary MECs (Figs. 2g–i and S5e–g). Mcam depletion significantly increased the numbers and diameters of

colonies, representing MEC clonogenicity and proliferation ability. Moreover, *Mcam* was knocked down using a lentivirus (Fig. S5h) to explore its function in primary cultured MECs in vivo and in vitro. In the transplantation assay, knockdown of *Mcam* significantly improved the regenerative capacity of MECs to reconstruct whole mammary tissue (Fig. S5i). In the clone assay, both clone numbers and diameters increased (Fig. S5j–m), suggesting loss of Mcam promotes MEC proliferation. These results provide strong evidences for the Mcam roles in inhibiting MEC proliferation and mammary reconstruction.

### Knockout of Mcam promotes macrophage recruitment

To identify the regulatory mechanism underlying the promotion function of MEC proliferation and mammary gland development in cKO mice by MMTV-Cre, RNA-sequencing (RNA-seq) was performed on mammary tissues from 5-week-old WT and cKO mice. First, we analyzed the differentially expressed genes (DEGs) identified between WT and cKO mammary tissues (total breast cells) (Fig. S6a) and performed Gene Ontology (GO) and Kyoto Encyclopedia of Genes and Genomes (KEGG) pathway enrichment analysis of upregulated DEGs in cKO mammary tissues. Results indicated strong activation of pathways associated with immune regulation (Fig. 3a). Notably, the DEGs were significantly enriched in the GO categories "immune system process" and "immune response" (Supplementary Dataset 2) and in the KEGG pathways "activation of innate immune response", "regulation of immune effector process", and "adaptive immune response" (Fig. 3a). Immune cells are key components of the mammary microenvironment[33], therefore, we analyzed the abundance of various immune cells in WT and cKO mammary tissues using FACS.

Data showed a significant increase in the abundances of macrophages and neutrophils in cKO mice, but not of dendritic cells (DC), B cells, or T cells (Figs. 3b and S6b). Furthermore, using macrophage markers (F4/80 and CD11b), macrophages showed consistent and strong elevation in the Mcam cKO mice (Figs. 3c and S6c), implying that macrophages play important roles in Mcam cKO mice. In addition to stromal macrophages, tissue-resident ductal macrophages are also found in the mammary gland epithelium[34]. Thus, we explored changes in both types of macrophages in the mammary gland using CD206 (stromal macrophage marker)[35] and Cx3cr1 (tissue-resident macrophage marker)[34]. Data showed that both cell populations were significantly elevated (Figs. 3d–g and S7a–d). To further validate macrophages indeed were recruited by MECs, isolated primary macrophages and macrophage cell line RAW264.7 were co-cultured with conditional medium (CM) from either WT or cKO MECs by MMTV-Cre. Compared with WT-CM treatment, cKO-CM induced more macrophages migration (Fig. 3h, i and S7e, f). Thus, depletion of Mcam in the mammary tissue leads to an increase in macrophages.

### Depletion of macrophages blocks the elevating activity of MECs upon Mcam loss

As macrophages can maintain MaSC stemness[36], we mixed macrophages with basal cells from WT or cKO mice by MMTV-Cre to implement a co-culture mammosphere assay for confirming Mcam roles on the clonogenicity promotion by macrophages. After co-culturing with isolated primary macrophages for 10-14 days, clone numbers and clone diameters increased significantly in the basal cells of cKO mice (Fig. 4a–c). To confirm the mediating roles of macrophages in the above function, we performed rescue assays using clodronate liposome (CL) to specifically deplete macrophages in both WT and cKO mice[37]. After injection of CL, the percentages of macrophages in WT and cKO mice were effectively reduced, as shown by FACS analysis (Figs. 4d and S8a), as was the percentages in the spleen (Fig. S8b, c), a macrophage-enriched organ.

We next determined the abundance of CD206[+] macrophages in the mammary tissues after CL treatment and found that the elevating CD206[+] signals upon Mcam loss were diminished completely (Fig. 4e,

f). Based on whole-mount staining, while Mcam cKO promoted mammary duct elongation within the fat pad (Fig. 1i), CL treatment eliminated these promotion effects (Figs. 4g, h and S8d). To further investigate the inhibitory effects of mammary morphogenesis after CL treatment, we determined the proliferation of MECs, from the IF (Fig. 4i, j) and IHC (Fig. S8e) staining results, both the elevating proliferation in basal and luminal cells by Mcam loss were flattened significantly after CL treatment, further indicating that amplification of both MEC lineages after Mcam loss was dependent on macrophages.

We next isolated and transplanted MECs from WT and cKO mice by MMTV-Cre into NOD/SCID mice to examine the repopulating ability of MECs. The MECs were mixed with CL to remove migrated macrophages, as performed in previous research[36]. While loss of Mcam increased the frequencies of MEC reconstitution (Fig. 2), we observed nearly complete inhibition of reconstitution in the fat pad when MECs were injected together with CL (Fig. 4k), suggesting that the promotion of MEC regenerative capacity upon Mcam cKO depends on the presence of macrophages. To further investigate the inhibitory effects of MaSCs after CL treatment, we determined the basal cell population percentages using FACS and found that systemic ablation of macrophages reversed the increase in Lin[-]CD24[+]CD29[+] cell abundance in cKO mice (Figs. 4l–m and S8f), suggesting that CL treatment rescued MaSC proliferation by Mcam loss. Taken together, these findings suggest that Mcam roles on clonogenicity, regeneration, and ductal morphogenesis of mammary glands are dependent on macrophages.

### Mcam recruits and activates macrophages through Il4-Stat6 axis

We next examined the molecular mechanism underpinning communications between macrophages and MECs. As macrophages were enriched by Mcam cKO (Fig. 3), we explored how the increased macrophages was recruited. Based on the RNA-seq DEGs (Fig. S6a), cytokines and chemokines increased in the cKO mammary tissues (Fig. 5a). The qRT-PCR results demonstrated that *Il4* was the most significantly elevated factors upon *Mcam* KO by both MMTV-Cre and K14-Cre (Figs. 5b and S9a). Il4 and Il13, which are secreted by MECs[38], and downstream Stat6 are essential for promoting macrophage polarization and activation[39,40]. Here, we found that Stat6 phosphorylation levels increased significantly in cKO mice (Figs. 5c and S9b), suggesting that Il4-Stat6 may mediate the role of Mcam in macrophages. To further validate this result, we blocked Il4 by its antibody in vitro and in vivo to examine macrophage migration. Blocking Il4 in vitro reversed the elevated macrophage migration upon Mcam loss (Fig. 5d, e). After injecting with Il4 blocking antibody orthotopically, we found the same rescue effects for CD206[+] and Cx3cr1[+] macrophages abundance (Fig. 5f–i), which leads to the inhibition of ductal elongation in Mcam cKO mice by MMTV-Cre ultimately (Fig. 5j, k). These data demonstrate Mcam roles for recruiting macrophages through Il4-Stat6 axis.

### Macrophages regulate MECs through non-canonical Wnt signaling upon Mcam loss

We also investigated the mechanism underlying the role of macrophages within mammary tissues. Previous reports have shown that macrophages play critical roles in MaSCs[36]. Based on gene set enrichment analysis (GSEA) of RNA-seq data, the gene cluster, canonical Wnt signaling pathway, showed no significant difference between WT and Mcam cKO mice by MMTV-Cre (Fig. S9c). Upon *Mcam* KO, all canonical Wnt ligands, including *Wnt1*, *Wnt3*, *Wnt3a*, *Wnt10a*, and *Wnt16*, were undetectable by qRT-PCR. For the downstream factors of canonical Wnt signaling, representative effectors, including *Axin2*, *GSK-3β*, and *β-catenin*, also showed no significant differences between WT and Mcam cKO mice (Fig. 6a). These data indicate that the promotion of MECs induced by Mcam deletion was not mediated by the canonical Wnt signaling pathway.

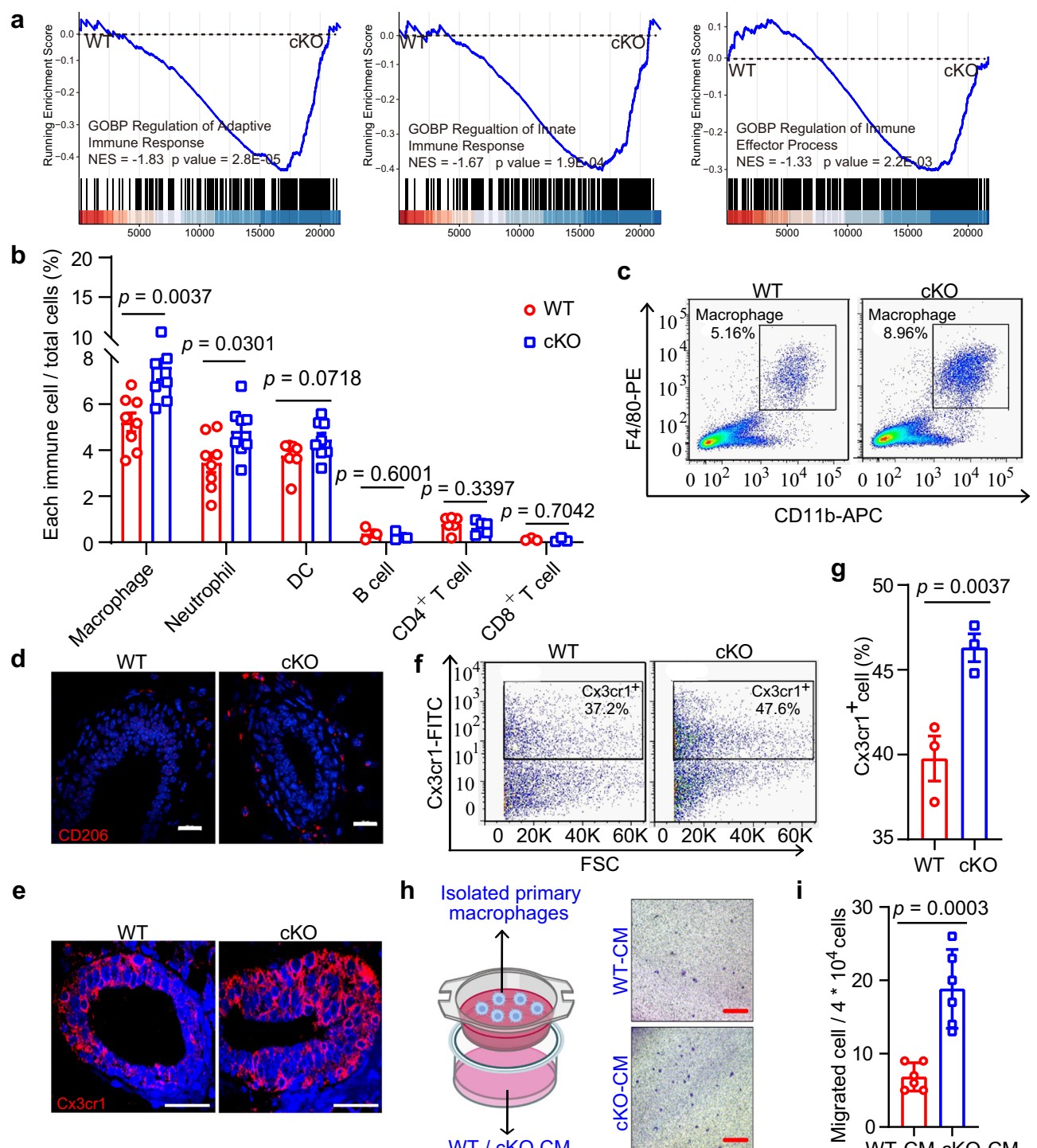

**Fig. 3 | Loss of Mcam promotes macrophage recruitment. a** GSEA of enriched immune response signatures using total mammary cells in cKO mice by MMTV-Cre compared with WT mice. NES, normalized enrichment score. **b** Percentages of various immune cell populations in WT and cKO mice by FACS analysis. n = 8 mice for macrophages, neutrophils, and DCs, 3 mice for B cells and CD8$^+$ T cells, and 6 mice for CD4$^+$ T cells. **c** FACS analysis of macrophage population (F4/80$^+$CD11b$^+$) in WT and cKO mice. **d**, **e** IF images of CD206$^+$ (**d**) and Cx3cr1$^+$ (**e**) macrophages in WT and cKO mammary sections. Scale bar, 20 μm. **f**, **g** FACS analysis (**f**) and their percentage quantification (**g**) of macrophages in mammary tissues of WT and cKO mice. *n* = 3 mice in each group. **h**, **i** Schematic of co-culture assay and representative images (**h**) (The left panel was created with BioRender.com.), and quantification (**i**) of migrating macrophages after WT and cKO CM treatment. Isolated macrophages by FACS in upper chamber were co-cultured with CM from WT and cKO primary MECs in lower chamber. CM, conditional medium. *n* = 6 biological replicates. Scale bar, 200 μm. Data are means ± SEM. Two-sided Student's *t* test was used to evaluate statistical significance. Source data are provided as a Source Data file.

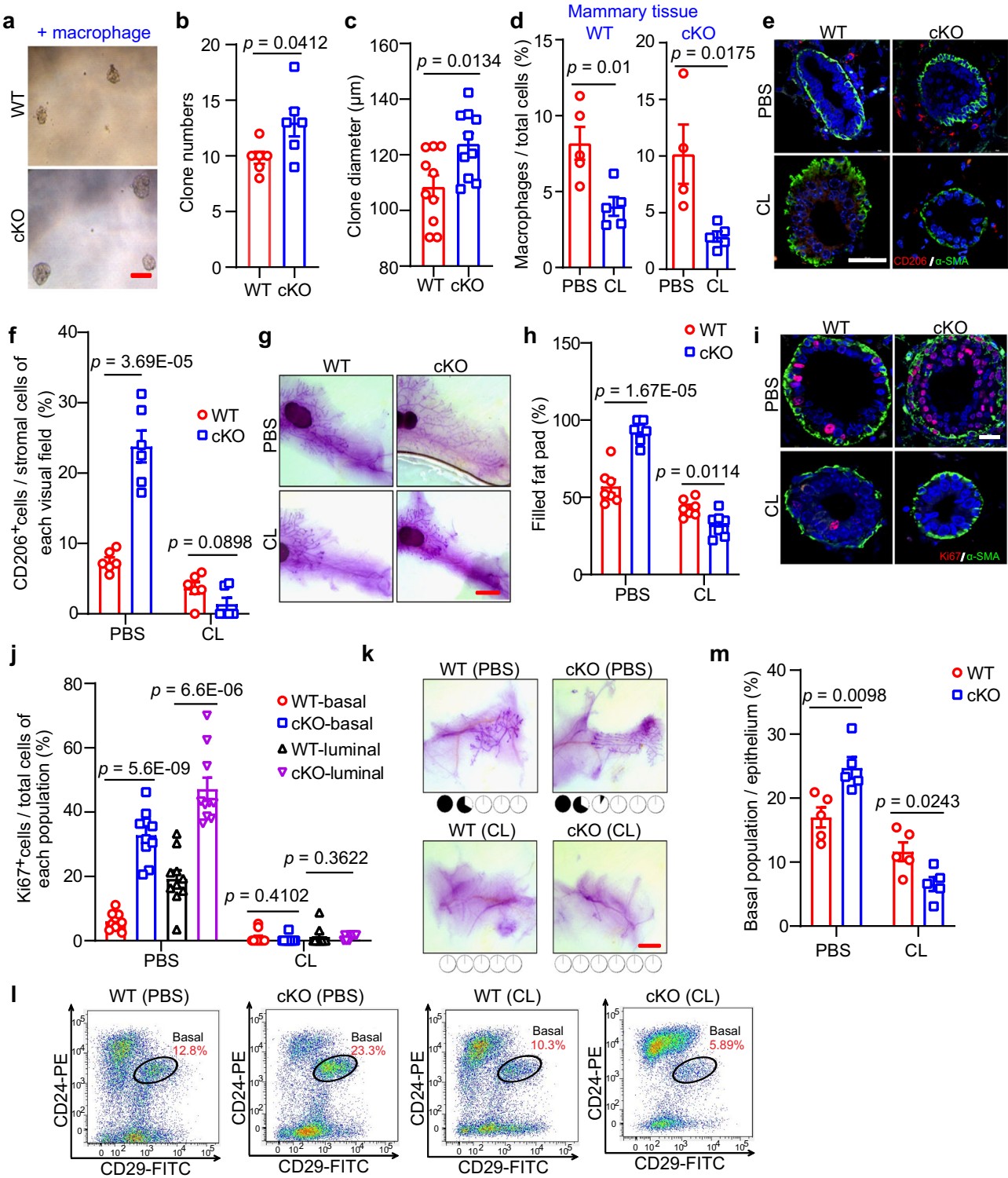

For non-canonical Wnt ligands, including *Wnt5a*, *Wnt5b*, and *Wnt11*, the level of *Wnt5a* was remarkable higher in the Mcam cKO group than in the WT group (Figs. 6a). This elevation of Wnt5a was verified at the protein level by IHC staining (Figs. 6b and S9d). Published data[21] have shown that Wnt5a is expressed in both basal and luminal epithelial cells. To confirm the Wnt5a surge in Mcam cKO mice is derived from macrophages, we isolated various cell populations in mammary tissues for measuring *Wnt5a* levels and taking *Wnt5a* expression in adipocyte of WT as a control, qRT-PCR showed macrophages (CD45+F4/80+) induced nearly 400-fold *Wnt5a* levels in cKO mice, and stromal cells containing macrophages also showed induced

*Wnt5a* levels in cKO mice (Figs. 6c and S9e). In addition, we depleted *Wnt5a* in macrophages (Fig. S9f) and transplanted these cells with basal cells to examine whether the absence of *Wnt5a* in macrophages could still enhance mammary development. While *Mcam* cKO promotes ductal elongation, loss of *Wnt5a* by shRNA in macrophages did not have the capacity to reconstruct whole mammary tissue (Fig. 6d), confirming Wnt5a is responsible for the phenotype observed.

We next tested whether Wnt5a mediates the maintenance of MEC clonogenicity and proliferation by adding Wnt5a blocking antibody to the culture medium in the coculture assay between cKO MECs and isolated macrophages. Results showed that in the absence of Wnt5a,

**Fig. 4 | Depletion of macrophages blocks the elevating clonogenicity and regenerative capacity of MECs upon Mcam loss. a–c** Representative images (**a**), clone numbers (**b**, 6 biological replicates in WT and 5 biological replicates in cKO by MMTV-Cre), and clone diameters (**c**, $n = 10$ in each group) of colonies formed by basal cells co-cultured with isolated macrophages. Scale bar, 100 μm. **d** Percentages of macrophages in mammary tissue of WT and cKO mice treated with clodronate liposome (CL) and their control groups (PBS). $n = 5$ mice in each group of WT. For cKO, $n = 4$ mice by PBS treatment, and $n = 5$ mice by CL treatment. **e, f** IF images of CD206[+] macrophages and α-SMA[+] basal cells (**e**) and quantification of CD206[+] macrophages in stromal cells (**f**) in WT and cKO mice with or without CL treatment. $n = 6$ sections in each group. Scale bar, 50 μm. **g, h** Representative images (**g**) and quantification (**h**) of the filled fat pad by the mammary duct of the whole-mount-stained mammary gland. Scale bar, 5 mm. $n = 7$ mice in each group. **i, j** IF images (**i**) and their quantification (**j**) of Ki67[+] proliferation cells and α-SMA[+] basal cells. $n = 10$ sections in each group. Scale bar, 20 μm. **k** Representative images of whole-mount-stained mammary outgrowths derived from transplantation of WT and cKO MECs mixed with PBS or CL. The number of MECs for transplantation was $1*10^4$. The dose of CL (90–125 μl) was determined according to the body weight (-9–12.5 g) of the mice, and PBS had the same volume as CL. Scale bar, 3 mm. **l, m** FACS analysis (**l**) and quantification (**m**) of basal population. $n = 5$ mice in each group. Mice used in figures **d–m** were treated with CL (150–170 μl) at 5 weeks of age (body weight -15–17 g) every other day for a week before harvesting the mammary gland. Data are means ± SEM. Two-sided Student's $t$ test was used to evaluate statistical significance. Source data are provided as a Source Data file.

Mcam loss failed to increase both clone numbers and clone diameters (Fig. 6e–g), indicating that Wnt5a mediates clonogenicity and proliferation promotion upon Mcam loss. To test the linkage between Il4 and Wnt5a, we blocked Il4 to prove that elevated Wnt5a expression levels are induced by Il4 stimulation. The results found the elevated Wnt5a levels were rescued by Il4 antibody injections (Fig. 6h, i), suggesting increased Il4 upon Mcam loss indeed activates Wnt5a expression.

### Loss of Mcam promotes MEC proliferation via Ryk

Next, we investigated how Wnt5a interacts with MECs. Wnt5a binds to two different receptors, i.e., Ror2 to inhibit branching morphogenesis and Ryk to promote MaSC/progenitor proliferation[23]. Thus, Ryk may be the potential receptor for our phenotype. Here, we explored the expression pattern of Ryk at the single-cell level using MEC single-cell RNA-seq from our and other laboratories. Based on the online data[29], Ryk was highly expressed in both basal and luminal cells, especially basal cells (Fig. 7a). Using a single-cell RNA-seq dataset generated by our laboratory[41], we found the same Ryk expression pattern (Fig. S10a). Interestingly, we analyzed the correlation between Mcam and Ryk, and found that Mcam and Ryk were co-expressed in many basal cells (Fig. 7b). This result was validated at the protein level using IF staining (Fig. 7c).

We next explored the expression levels of the two receptors using qRT-PCR. Results showed that upon Mcam loss *Ror2* was decreased and *Ryk* was elevated (Fig. 7d), especially in basal cells (Fig. 7e), consistent with the promoting role of Ryk in MaSCs found in previous study[23]. The increase in Ryk expression upon Mcam cKO was confirmed by WB or IHC (Figs. 7f and S10b). Moreover, Wnt5a blocking resulted in a decrease of Ryk expression, whereas exogenous recombinant Wnt5a (rWnt5a) increased Ryk expression (Fig. S10c, d), which is consistent with the conclusion that Ryk is correlated with the levels of its ligand - Wnt5a[42–44]. These results imply that the phenotypes observed in the current study is mediated by the Ryk receptor. To further test this hypothesis, we blocked Ryk using its antibody and performed a rescue assay. Results showed that both the increased clone numbers and clone diameters caused by Mcam loss were suppressed to a level similar to that of WT after blocking Ryk (Fig. 7g–i), suggesting that the Mcam functions on MECs is dependent on Ryk. Taken together, loss of Mcam leads to an increase of Wnt5a, which binds to the Ryk receptor, thereby contributing to the clonogenicity, regenerative capacity, and proliferation of MECs.

### Discussion

Both canonical (β-catenin dependent) and non-canonical (β-catenin independent) Wnt pathways regulate the maintenance, self-renewal, and differentiation of adult stem cells[45]. Canonical Wnt signaling tends to be involved in stem cell self-renewal and progenitor cell proliferation and differentiation[45,46], whereas non-canonical Wnt signaling tends to be involved in stem cell maintenance and directional cell movement[4], but can indirectly regulate adult stem cells by inhibiting canonical Wnt signaling[7]. In our study, Mcam acted as a co-receptor of the non-canonical Wnt signaling pathway to inhibit regenerative capacity and proliferation of MEC independent of canonical Wnt signaling, providing direct evidence that non-canonical Wnt signaling can regulate MaSCs independent of canonical Wnt signaling. Previously, several studies showed Ryk was correlated with the levels of its ligand - Wnt5a[42–44]. In our study, Wnt5a blocking resulted in a decrease of Ryk expression, and exogenous recombinant Wnt5a (rWnt5a) increased Ryk expression (Fig. S10c, d). These data speculates the mediating role of Wnt5a for potential balance between the levels of Mcam and Ryk, which need to be investigated in the future.

Cell-extrinsic signals from the microenvironment control stem cell maintenance and regulate their function in adult tissues. Macrophages in the mammary gland microenvironment (stromal macrophages)[47] or within mammary ducts (tissue-resident macrophages)[36] dictate stem cell function, including self-renewal activity. However, the underlying mechanisms remain poorly understood. In the current study, we found that the role of macrophages in promoting MECs was achieved via the Il4-Stat6 axis and Wnt5a/Ryk-mediated non-canonical Wnt signaling pathway. This study on normal mammary gland development may provide important clues about the roles of macrophages in mammary tumorigenesis.

Positive regulators of MECs have been studied extensively due to their potential as therapeutic targets for breast cancer[1,48–50], with many such regulators found to function through the canonical Wnt signaling pathway[2,50,51]. However, an increasing number of negative regulators of MECs have been identified, including Prox1, Asap1[52], Wnt5b[53], Sirt4[54], Becn1[55], and Adam17[56], and their potential functions have garnered increasing attentions. In addition, although the role of canonical Wnt signaling has been explored, the role of non-canonical Wnt signaling in MaSCs is not entirely clear. Our study presents Mcam, a highly expressed factor in MECs, which acts as a receptor of non-canonical Wnt5a to negatively regulate MECs and mammary gland development. These findings further support the important roles of negative regulators in mammary gland development.

Published data[21,29] have shown that Wnt5a is expressed in both basal and luminal epithelial cells, with a higher level in luminal cells. Indeed, in our data Wnt5a expressed in luminal cells higher than basal cells (Fig. 6c), but no significant difference between two cell layers upon Mcam loss (Fig. 6c). In comparison with attentions for Wnt5a in MECs, the expression profile of Wnt5a in mammary microenvironment including stromal cells and macrophages is less clear, our data demonstrated that macrophages are the main origin of Wnt5a, especially upon Mcam loss (Fig. 6), emphasizing correlation between non-canonical Wnt signaling and macrophages.

In our data, we found heterozygous loss of Mcam by K14-Cre (Mcam[fl/wt]/K14-Cre) had similar phenotypes to homozygous loss by MMTV-Cre mice. This may be because the MMTV promoter is chimerically activated in mammary epithelial cells[31], while K14 promoter is specifically activated in basal cells[57] and much more robust. Moreover, Mcam is predominantly expressed in the basal layer of the mammary

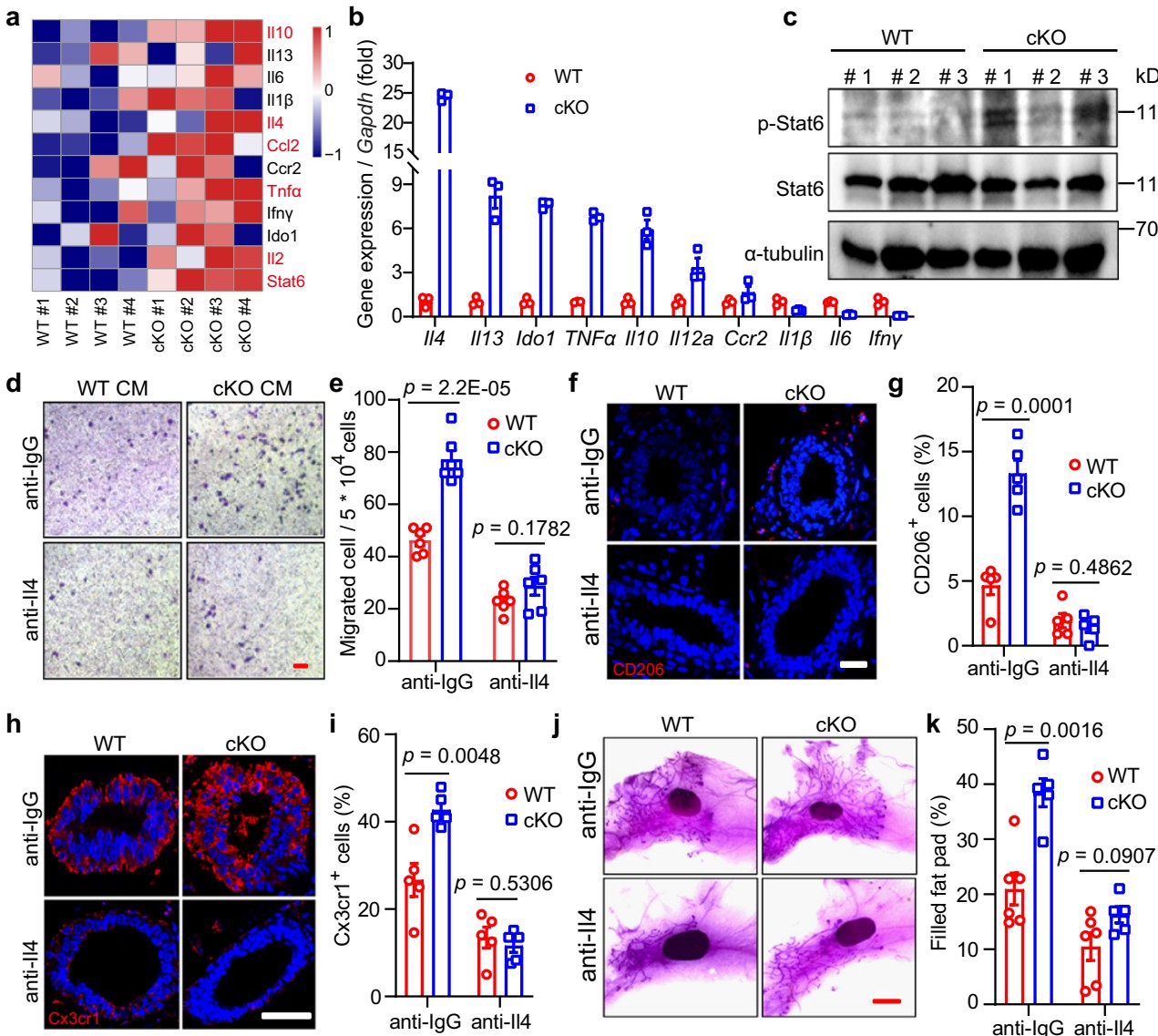

**Fig. 5 | Mcam loss recruits macrophages through Il4-Stat6 axis. a** Heat map showing differentially expressed cytokine genes based on RNA-seq data in cKO mice by MMTV-Cre compared with WT mice. Differentialy expressed genes are labelled in red. **b** qRT-PCR analysis of expression levels of cytokines in MECs from WT and cKO mice. $n = 3$ biological replicates in each group. **c** Western blotting showing Stat6 and p-Stat6 levels in WT and cKO mammary tissues. $n = 3$ mice in each groups. **d, e** Representative images (**d**) and their quantification (**e**) of migrated primary macrophages incubated with Il4 antibodies or their control (anti-IgG) after treatment with conditional medium (CM) of WT or cKO MECs. n = 6 biological replicates in each group. Scale bar, 100 μm. **f, g** IF images of CD206⁺ macrophages (**f**) and their statistical results of CD206⁺ macrophages (**g**) in WT and *Mcam* cKO mammary tissues treated with IgG or Il4 antibodies. WT and cKO Mice were treated with IgG (10 μl) or Il4 antibodies (10 μl) orthotopically at 5 weeks of age every other

day for a week before harvesting the mammary tissue. Scale bar, 20 μm. n = 3 mice were treated for each groups. 5 fields in each sample for statistics. **h, i** IF images of Cx3cr1⁺ macrophages (**h**) and their statistical results (**i**) in WT and Mcam cKO mammary tissues treated with IgG or Il4 antibodies. Scale bar, 50 μm. 3 mice were treated in each group. 5 fields in each sample for statistics. **j, k** Representative images (**j**) and quantification of filled fat pad (**k**) of whole-mount-stained mammary tissues in WT and cKO mice treated with IgG or Il4 antibodies. $n = 6$ in WT group and $n = 5$ in cKO group. Scale bar, 5 mm. WT and cKO Mice in Figure **h–k** were treated with IgG (10 μl) or Il4 antibodies (10 μl) orthotopically at 5 weeks of age every other day for a week before harvesting the mammary gland. Data are means ± SEM. Two-sided Student's *t* test was used to evaluate statistical significance. Source data are provided as a Source Data file.

gland, the phenotypes using K14-Cre mice were therefore much more intense than those by MMTV-Cre mice. Actually, we found the above-mentioned phenotypies had the similar protein levels of Mcam (approximately 30% to their controls) in total breast tissues (Fig. 1h, S3e, and S10e-S10f). Meanwhile, in particular basal layer, we also detected the expression profile of these two genotypes, IF staining showed the similar Mcam depletion between heterozygous and homozygous knockout of Mcam (Figs. 1g and S3d). As to why heterozygous loss of Mcam has the similar expression profile to homozygous loss, this could be due to the expression bias[58] of floxed allele of

Mcam. In previous reports, heterozygous loss of Brca1 or Tet2 had less than 10% expression levels of their control groups, which were very close to the homozygous loss of these genes[32,59], demonstrating the expression bias of these floxed alleles.

In conclusion, we found that Mcam was highly expressed in basal cells and showed inhibitory effects on MEC proliferation and mammary ductal morphogenesis. Depletion of Mcam promoted the recruitment and activation of macrophages via the Il4-Stat6 axis and promoted macrophage secretion of Wnt5a. In addition, the loss of Mcam resulted in a marked increase in the non-canonical Wnt receptor

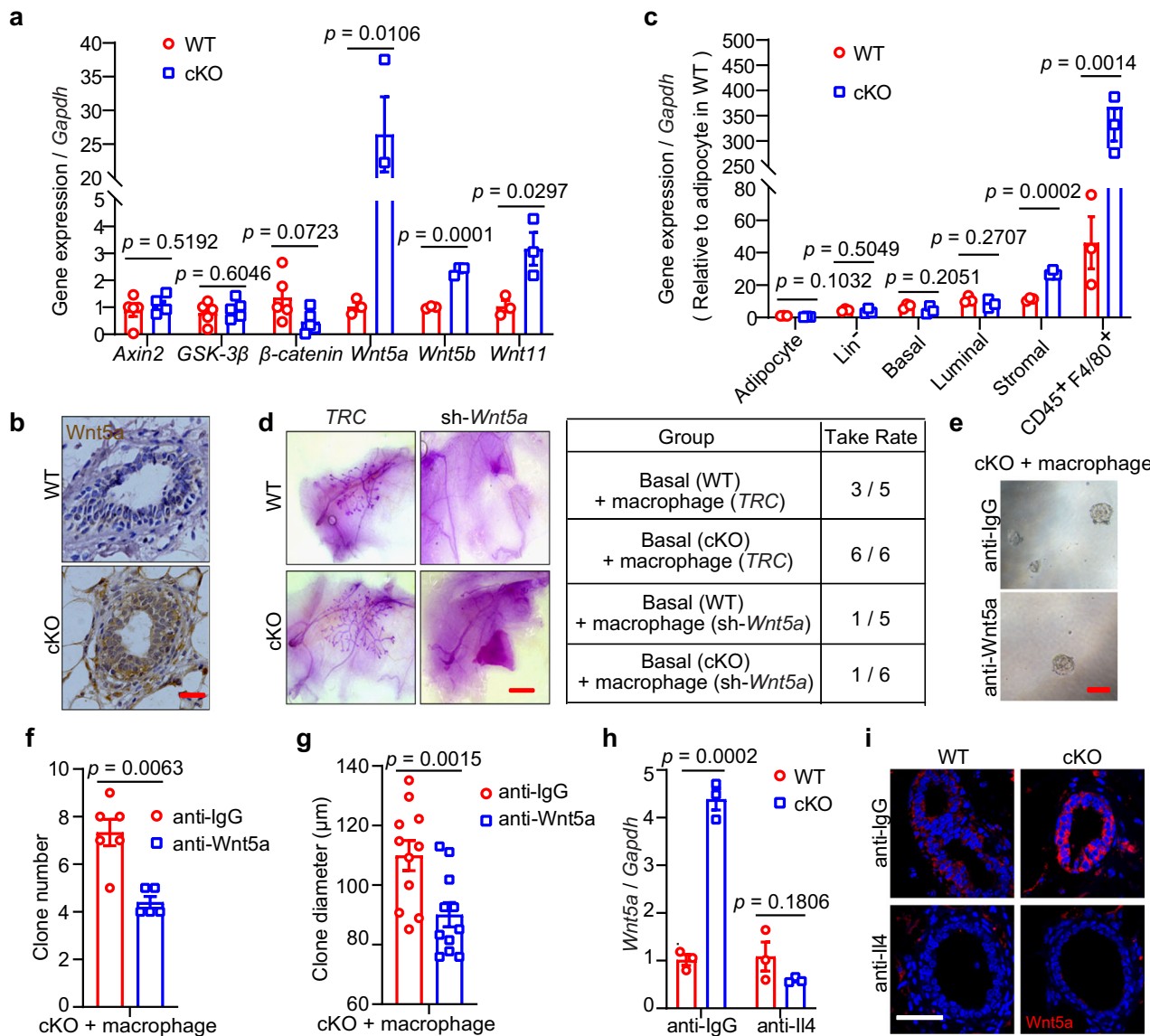

**Fig. 6 | Macrophages regulate MECs through Wnt5a upon Mcam loss. a** qRT-PCR analysis of *Axin2, GSK-3β, β-catenin* (n = 5 mice in each group), *Wnt5a, Wnt5b,* and *Wnt11* (n = 3 mice in each group) expression in mammary tissues from WT and cKO mice by MMTV-Cre. **b** IHC staining of Wnt5a in WT and cKO mice. Scale bar, 20 μm. **c** qRT-PCR results for *Wnt5a* in indicated cell populations of breast tissues. *Wnt5a* expressed in adipocyte of WT was taken as a control. The results are from 3 independent mice. **d** Representative images (left) and reconstitution efficiency (right) of whole-mount-stained mammary outgrowths derived from transplantation of isolating mammary macrophages transducing with sh*Wnt5a* together with WT or cKO basal cells. 200 basal cells and 800 isolated macrophages were transplanted into cleared mammary fat pads of recipient mice. n = 5 for WT group and n = 6 for cKO group. Scale bar, 5 mm. **e–g** Representative images (**e**), clone numbers (**f**, n = 6 for anti-IgG group and n = 5 for anti-Wnt5a group), and clone diameter (**g**, n = 11 for each group) of colonies formed by cKO mammry cells cocultured with isolated macrophages treated with anti-IgG or anti-Wnt5a, respectively. 200 mammary cells and 800 isolated macrophages were seeded in each group. Scale bar, 100 μm. **h** qRT-PCR analysis of expression levels of *Wnt5a* in WT and cKO mammary cells cocultured with isolated macrophages treated with Il4 or IgG antibodies. n = 3 replicates for each group. **i** IF staining of Wnt5a in mammary tissues of WT and cKO mice after treatment with Il4 antibody in vivo. Scale bar, 50 μm. Data are means ± SEM. Two-sided Student's *t* test was used to evaluate statistical significance. Source data are provided as a Source Data file.

Ryk in basal cells. This process drove Wnt5a and Ryk binding, which promoted excessive MEC proliferation, which may stimulate mammary tumorigenesis (Fig. 8).

## Methods

### Mice

C57/B6 mice (8-week-old) were purchased from the Experimental Animal Center of the Kunming Institute of Zoology (China). The Mcam floxed mice (8-week-old) have been described previously[30]. The MMTV-Cre (line D) and K14-Cre mice (8-week-old) were obtained from the Jackson Laboratory (USA). The female NOD/SCID mice

(3-week-old) were purchased from the company of Charles River. The mice were raised in a specific pathogen free (SPF) environment with an ambient temperature of 18-22 °C, a humidity of 50%-60%, and a 12 h light-dark cycle. All experimental procedures and animal care and handling were performed according to the protocols approved by the Ethics Committee of the Kunming Institute of Zoology, Chinese Academy of Sciences (IACUC-RE-2023-07-006).

### Mammary epithelial cell preparation and culture

Mammary glands from 8–10-week-old C57/B6 female mice were minced and digested in Dulbecco's Modified Eagle's Medium/Nutrient

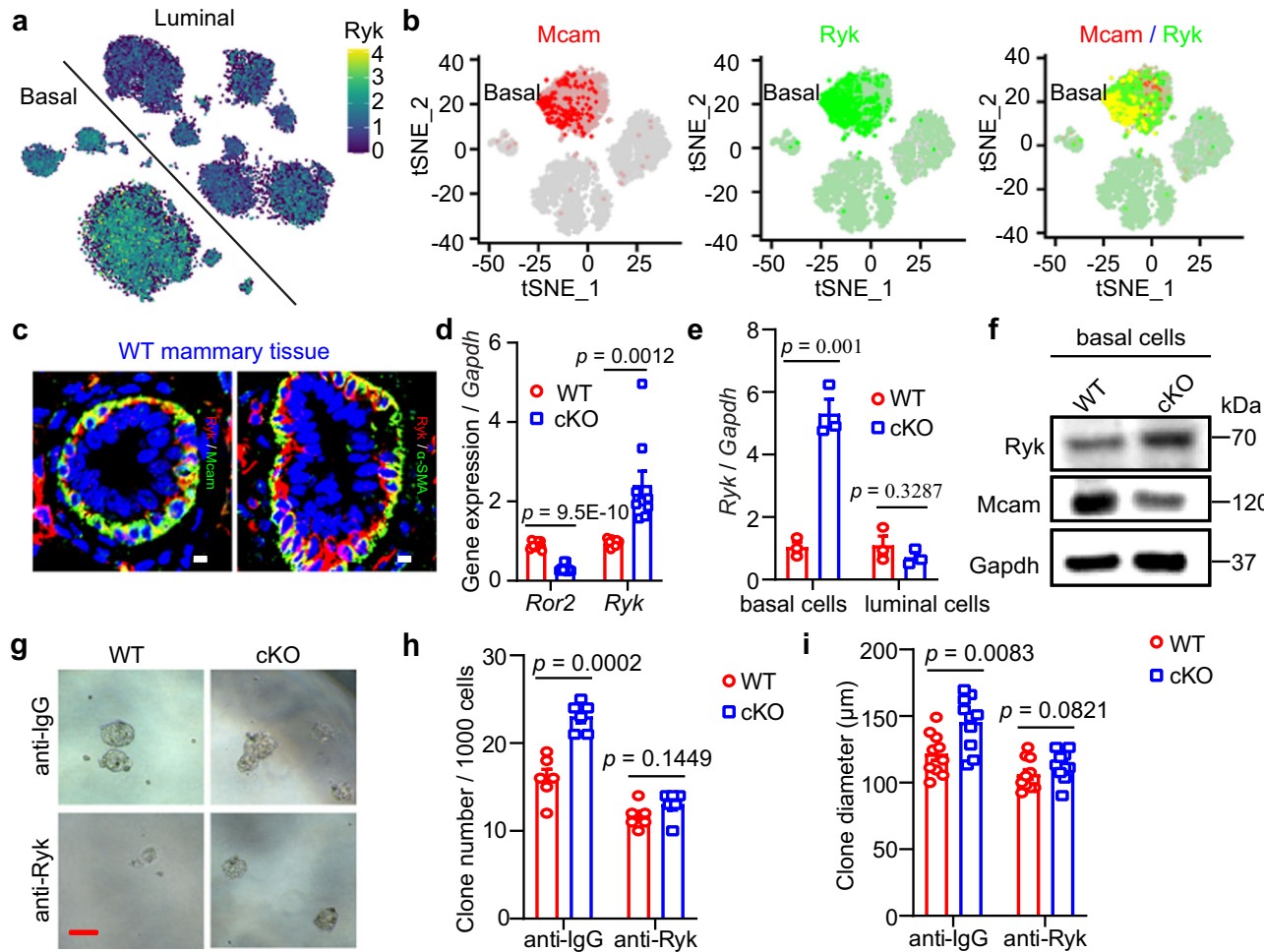

**Fig. 7 | Loss of Mcam promotes MEC proliferation via Ryk. a** tSNE plot showing expression pattern of Ryk in different lineages based on single-cell sequencing data[29]. **b** tSNE plot showing expression pattern of Mcam (red), Ryk (green), and overlap (orange) in basal cells based on single-cell sequencing data[41]. **c** IF images of co-staining of indicated antibodies in WT mammary tissues. Scale bar, 5 μm. **d** qRT-PCR analysis of *Ror2* and *Ryk* in MECs. *n* = 9 mice in each group. **e** *Ryk* expression levels in basal and luminal cells isolated from WT and cKO mammary tissues by MMTV-Cre. *n* = 3 biological replicates. **f** Western blotting showing Ryk protein expression in basal cells from WT and cKO mice. The experiment was repeated three times independently. **g**–**i** Representative images (**g**), clone numbers (**h**, *n* = 6 replicates for each group), and clone diameter (**i**, *n* = 10 replicates for each group) of colonies formed by WT and cKO mammry cells treated with anti-IgG or anti-Ryk, respectively. Scale bar, 100 μm. Data are means ± SEM. Two-sided Student's *t* test was used to evaluate statistical significance. Source data are provided as a Source Data file.

Mixture F-12 (DMEM/F12) containing 5% fetal bovine serum (FBS), 1% penicillin-streptomycin, 300 U/ml collagenase I (Sigma, C0130), and 100 U/ml hyaluronidase (Sigma, H3506) for 1–2 h at 37 °C. Single-cell suspension was obtained by sequential incubation with 0.25% trypsin EDTA for 5 min and 5 mg/ml dispase (Sigma, D4693) containing 0.1 mg/ml Dnase I (Roche, 11248932001) for 5 min at 37 °C with gentle shaking. Finally, red blood cells were removed with 0.8% $NH_4Cl$, and the cell suspension was then filtered through a 40-μm cell strainer. Primary MECs were cultured in DMEM/F12 with 2% NBCS, 10 ng/ml EGF, 4 μg/ml heparin, 5 μM Y-27632, 10 ng/ml bFGF, and 0.5 μM BIO.

### Pooled library transduction into progenitor-enriched cells

After obtaining progenitor-enriched cells (clones derived from clone assay), the virus was titered by suspension of $1 \times 10^5$ cells/well in an ultra-low 6-well plate with different virus dilutions in each well for 24 h at 37 °C. Two replicates were used for each dilution concentration. At 24 h post-transduction, 0.8 μg/ml puromycin was added to one of the replicate wells. After 48 h, cells were counted in all wells to determine the optimal MOI. Detailed methods are discussed in previous research[27].

A total of $1.25 \times 10^7$ cells (nearly 100 × coverage) were infected at the optimal MOI and selected with 0.8 μg/ml puromycin for 7 days and split every 2–3 days. After 7 days, $4 \times 10^6$ cells (nearly 40 × coverage) were spun down and frozen for genomic DNA extraction. In addition, $4 \times 10^6$ cells were used to carry out the clone assay and a similar number of cells derived from large clones were used for the transplantation assay. At the end of the experiment, large clones or regenerated tissues were collected and frozen for genomic DNA extraction.

### Genomic DNA extraction from cells, large clones, and regenerated tissues

Genomic DNA was extracted using a gDNA extraction kit (Axygen, USA) following the procedure instructions provided. The gDNA concentration was measured using a NanoDrop spectrophotometer.

### sgRNA library readout by deep sequencing

The sgRNA library readout was performed using a two-step PCR procedure, with the first PCR step including enough genomic DNA to preserve full library complexity and the second PCR step adding appropriate sequencing adapters to the products from the first PCR.

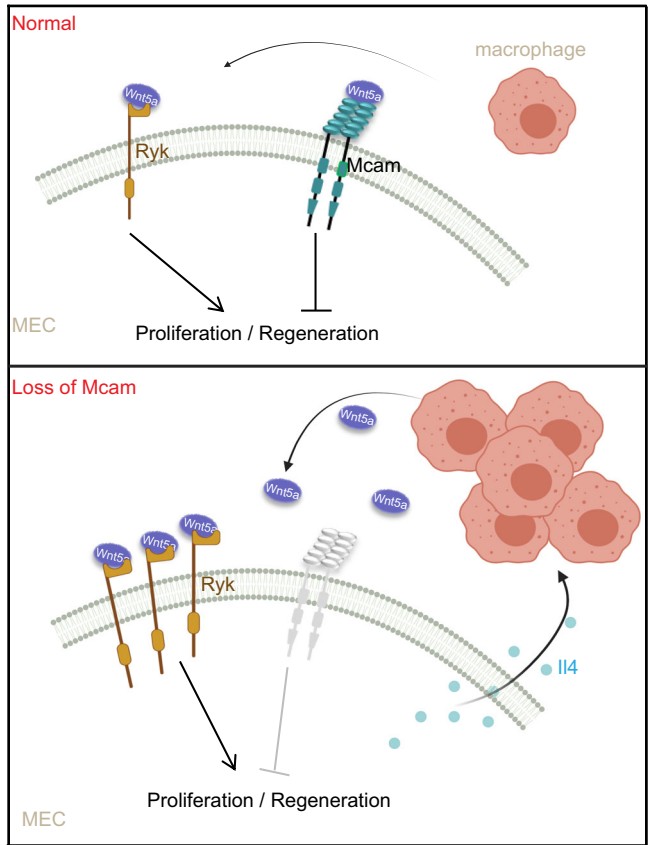

**Fig. 8 | Working model.** In basal cells of normal mammary gland epithelium, Mcam inhibits, and Ryk promotes MEC proliferation to sustain the homeostasis of mammary gland development. Loss of Mcam promotes the recruitment and activation of macrophages via the Il4-Stat6 axis and thereafter induces Wnt5a secretion by macrophages. The binding of Wnt5a to Ryk, a non-canonical Wnt receptor, elevates regenerative capacity and excessive MEC proliferation. Created with BioRender.com.

The sgRNA library for each sample (plasmids, genomic DNA from cells, large clones, and regenerated tissues) was amplified and prepared for Illumina sequencing using the following two-step PCR procedure. For PCR#1, a region containing sgRNA cassette was amplified using primers specific to the sgRNA-expression vector (lentiCRISPRv2-PCR1-F and lentiCRISPRv2-PCR1-R): lentiCRISPRv2-PCR1-F: GAAAGTAA-TAATTTCTTGGGTAGTTT; lentiCRISPRv2-PCR1-R: GTCCAC-CAGTTTCTTTCTCAGGT. For PCR#1, 600 ng of gDNA was used, with thermocycling parameters: 98 °C for 30 s, 21 cycles of (98 °C for 10 s, 55 °C for 10 s, 72 °C for 30 s), and 72 °C for 2 min. The PCR#1 products for each biological sample were pooled and used for amplification with barcoded second PCR primers (Supplementary Dataset 3). The second PCR products were pooled and gel purified. The purified pooled library was then quantified with dsDNA High-Sensitivity Qubit Kit (Life Technologies, USA). The samples were then sequenced using Illumina X-Ten.

### sgRNA deep sequencing data processing
Deep sequencing data were processed for sgRNA representation using custom scripts. In the double-terminal data produced by sequencing, one read end contained sgRNA, and the other end was sequenced on the vector. Firstly, segments that did not contain sgRNAs were removed using Cutadapt v2.1 (DOI:10.14806/ej.17.1.200) and specific primer sequences. Secondly, due to poor polymorphism of the sequenced samples, many sgRNA sequences in the sequenced reads exhibited positional shifts (theoretically, sgRNA should start at 23 bp in the sequenced reads, but may move back and forth by 1–10 bp). If 23–42 bp is directly extracted as sgRNA by Cutadapt, it can lead to the

loss of sgRNA. Therefore, we used the sequencing library as a reference index and the mouse GeCKO v2 library as the input reading section. We then used a self-programmed Python script to extract the sequencing depth of each sgRNA from the BAM alignment file. If the sgRNA sequence from the GeCKO v2 library can be aligned to the 15–50 bp of some reads and there is no insertion or deletion information in the alignment region of sgRNA, At the same time, there is no more than one mismatch base between the alignment sequence of sgRNA and the sequencing read segment, then the region of the corresponding sequencing read segment is the real sgRNA sequence, and the number of this sgRNA is increased by 1. Finally, the sgRNA count matrix was generated, i.e., the sequencing depth of each sgRNA in each sample.

A large quantity of sgRNAs were transduced into the mammary cells, resulting in their enrichment through the formation of colonies. As stated, the following criteria were used for selecting sgRNAs: (i) If the sgRNA failed to induce any alterations in its targeted gene and colonies either did not form or remained small in size, then it was not considered for further analysis. (ii) If the sgRNA induced alterations in the targeted gene but the gene did not significantly impact clonal growth, it was not selected as it shared the same characteristics as described in section (i). (iii) If the sgRNA induced alterations in the targeted gene, and the loss of that gene promoted or enabled clonal growth, it was considered enriched and was subjected to deep sequencing.

Statistical analysis of sgRNA enrichment was performed using MAGeCK software (http://bitbucket.org/liulab/mageck-vispr)[28]. The sgRNA count matrix was converted into the input format recognized by MAGeCK, and the MAGeCK test command was run for statistical analysis to obtain a list of significantly enriched sgRNAs. Enriched sgRNAs were detected for each mammary tissue and large clone compared to small clones or to the cell library. Candidate sgRNAs with possible biological effects were selected based on the following conditions: (1) at least two sgRNA enriched in a large clone sample than cell library, (2) at least two sgRNA enriched in a large clone sample than a small clone sample, (3) at least two sgRNA enriched in all regenerated tissues (except sample named S5979, which had poor sgRNA library quality) than cell library, (4) the targeted gene of enriched sgRNAs are showed to be highly expressed in basal cells than luminal cells.

### Plasmid construction and knockdown
For knockdown, shRNAs were cloned into the pLKO.1 (AddGene #8453) vector. The shRNA sequences are listed in Supplementary Dataset 3. Sequencing verified all plasmid constructs to exclude mutations. These vectors were co-transfected with psPAX2 (AddGene #12260) and pMD2.G (AddGene #12259) (4:3:1) into HEK293T cells (Conservation Genetics CAS Kunming Cell Bank #KCB 200744YJ) to produce lentiviral particles, which were then transfected into MECs. After 72–96 h, the cells were collected for further analysis. We used sh-TRC (Addgene #10879) as the shRNA control. All primers used in this study are listed in Supplementary Dataset 3.

### RNA extraction and qRT-PCR
Total RNA was extracted using TRIzol reagent (Life Technologies) and then converted to complementary DNA using a PrimeScript™ RT Reagent Kit (TaKaRa, containing gDNA Eraser). We then performed qRT-PCR on a QuantStudio 3 instrument using a SYBR Green PCR Master Mix (Applied Biosystems). Primers used are listed in Supplementary Dataset 3.

### Western blot analysis
Protein lysates were electrophoresed by sodium dodecyl sulfate (SDS)-polyacrylamide gel electrophoresis and transferred to polyvinylidene difluoride membranes. Blots were incubated in 5% non-fat dry milk for 1 h and primary antibodies at 4 °C overnight, and then incubated with horseradish peroxidase (HRP)-linked secondary antibodies (Sigma) for

1 h at room temperature. Protein expression was detected using a chemiluminescent HRP substrate (Millipore). Western blot images were taken using SageCapture software. All antibodies used in this study are listed in Supplementary Dataset 4. Source data are provided as a Source Data file.

## Clone assay

For the clone assay, MECs were plated in ultra-low adherence plates (Corning) in clone assay medium consisting of EpiCult-B Medium (Mouse) kit (STEMCELL Technologies, 05610), 5% FBS, 5% Matrigel, 10 ng/μl EGF, 20 ng/μl FGF, heparin, and Y-27632. The medium was changed every 3–4 days, and the number and diameter of clones were counted after culturing for 7–14 days at 37 °C. For the co-culture clone assay, 200 MECs were mixed with 800 macrophages and grown on ultra-low adherence plates (Corning) in clone assay medium. Procedures were performed according to previous work[20].

## Transplantation assay

Cell were manually counted and transplanted at limiting dilution, as described previously[24], in the presence of Matrigel. Cells were resuspended in 50% Matrigel and 0.04% Trypan Blue and injected at 10-μl volumes into cleared fat pads of 3-week-old female NOD/SCID mice. Reconstituted mammary glands were harvested at 5 or 6 weeks post-surgery. For the transplantation assay with CL treatment, protocols followed previously published research[36]. For transplantating mixed basal cells and macrophages, 200 basal cells were mixed with 800 macrophages.

## Flow cytometry

The sorting of basal and luminal cell populations followed our previously published work[41]. For analysis of T cells, B cells, dendritic cells (DCs), neutrophils, and macrophages, cells prepared from the mammary gland and spleen were stained using a subset of antibodies. CD3-APC and CD4-FITC were used for sorting CD4+ T cells, CD3-APC and CD8-PE were used for sorting CD8+ T cells, CD11b-APC, CD45-FITC and F4/80-PE were used for sorting macrophages, CD11b-APC and Ly6G-FITC were used for sorting neutrophils, B220-FITC and CD19-PE were used for sorting B cells, and CD11c-PB and MHC-II were used for sorting DCs. All antibodies are listed in Supplementary Dataset 4.

## Clodronate liposome assay

CLs are nontoxic until ingested by macrophages, after which they are broken down by liposomal phospholipases to release the drug that subsequently induces macrophage death by apoptosis[37]. Detailed methods of CL assay have been reported in previous research[20]. For systemic treatment, Mcam cKO mice were treated with CL (130–150 μl) at 4 weeks of age (body weight -13–15 g) every other day for a week before the mammary glands were harvested. For the transplantation assay, MECs from WT and Mcam cKO mice were mixed with or without CL following previously published procedures[36] and then injected into the cleared mammary fat pad of NOD/SCID mice. Transplants were harvested at 5 weeks post-injection.

## Transwell assay

Briefly, MECs from WT and Mcam cKO mice were cultured for 24 h, then cultured with basic medium (without serum) for another 24 h at room temperature. This conditional medium (CM) was then collected. Isolated macrophages or macrophage cell line RAW264.7 (a gift from Xiaopeng Qi lab, Shandong University) were plated in the upper chamber and CM was added in the lower chamber. After culturing for 6 h, the migrated macrophages were counted.

## Whole-mount staining

The fourth mammary glands were excised and spread on microscope slides and fixed in 25% glacial acetic acid and 75% ethanol for 1 h.

Tissues were then stained with carmine alum solution overnight at 4 °C. After staining, the slides were dehydrated through increasing ethanol concentrations, cleared in xylene, and coverslipped with Neutral Balsam (Solarbio, G8590).

## Histology and immunostaining

Mammary tissues were fixed in formalin and embedded in paraffin. Tissue blocks were then sectioned (5-μm thickness) and stained with hematoxylin and eosin (H&E). For immunostaining, sections were dehydrated with graded alcohol and boiled in 10 mM sodium citrate for antigen retrieval for 20 min. Sections were then used for IHC and IF analyses. For immunohistochemistry, sections were incubated three times with 3% $H_2O_2$ for 5 min at room temperature to inactivate endogenous peroxidases, and then blocked with 10% goat serum for 2 h and incubated with primary antibodies at 4 °C overnight. The slides were then washed three times in phosphate-buffered saline (PBS) and incubated with secondary antibodies for 1 h at room temperature and developed with 3,3'-diaminobenzidine. For IF, sections were blocked for 2 h, and then incubated with primary antibodies for 2 h and secondary antibodies for 1 h at room temperature after direct antigen retrieval. The slides were counterstained with 4',6-diamidino-2-phenylindole dihydrochloride (Vectashield, H-1200, Vector Laboratories). Microscopy images were taken using NIS-Elements F 4.0 software, and analyzed using Image-Pro Plus 5. All antibodies are listed in Supplementary Dataset 5.

## Blocking Il4 in vivo orthotopically

For blocking Il4 in vivo, Il4 or IgG antibody were injected into WT and cKO mammary gland fat pad. Mammary tissues were harvested at 1 week post-injection and carried out whole-mount staining. The left mammary gland of 4-week-old were treatd with IgG antibody (100 μg) and the right with Il4 antibody (100 μg).

## RNA-seq analysis

Raw sequencing data were processed through standard Illumina pipelines for base-calling and fastq file generation. Paired-end reads were mapped to the mouse genome primary assembly (NCBIM37) and the Ensembl mouse gene annotation for NCBIM37 genebuild was used to improve mapping accuracy with STAR v2.4.2a[60]. FeatureCounts v1.4.6-p5[61] was used to assign sequence reads to genes. Mitochondrial genes, ribosomal genes, and genes possessing less than five raw reads in half the samples were removed. Differential expression analysis was performed with the Bioconductor edgeR package v1.6[62]. Significant genes were determined by an adjusted P-value of <0.01 based on Benjamini-Hochberg multiple testing correction and log-2-transformed fold-change of > 1 or <− 1. After obtaining the list of DEGs, the genes were analyzed using Gene Ontology to identify enriched pathways and biological processes.

## Reporting summary

Further information on research design is available in the Nature Portfolio Reporting Summary linked to this article.

## Data availability

The raw sequence data generated in this study have been deposited in the Genome Sequence Archive[63] in National Genomics Data Center[64], China National Center for Bioinformation / Beijing Institute of Genomics, Chinese Academy of Sciences that are publicly accessible at https://ngdc.cncb.ac.cn/gsa/. The accession codes are GSA: CRA011830, GSA: CRA011840. The source data underlying Figs. 1c, f, h, j, l, 2b, d, h, i, 3b, g, i, 4b–d, f, h, j, m, 5b, c, e, g, i, k, 6a, c, e–h, 7d, e, h, i and Supplementary Figs. 3b, c, e, g, j, 4e, g, b, c, f–h, k–m, 7f, 8c, 9a, b, 9f, 10c−f are provided as a Source Data file. Source data are provided with this paper.

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

## Acknowledgements

This work was supported by the National Key Research and Development Program of China (2020YFA0112300, B. J.), National Natural Science Foundation of China (U23A20452, B. J., and 32160174, Xing Y.), Yunnan Applied Basic Research Key Projects (202201AT070279, Xing Y., 202101AU070125, Xing Y.), Yunnan Science and Technology Department-Kunming Medical University Joint Fund for Applied Basic Research (202101AY070001-212, Xing Y.), Spring City Project from Kunming Science and Technology Bureau (2022SCP007), and Open subject of Key Laboratory of Tumor Immunological Prevention and Treatment of Yunnan Province (KLTIPT-2023-08, Xing Y.). N.Y.F. was supported by Singapore NMRC OF-IRG (MOH-OFIRG20nov-0018) and Victorian Cancer Agency Mid-career Research Fellowship. We thank Dr. Christine Watts for English editing and Guolan Ma and Shuangjuan Yang from the Core Technology Facility, Kunming Institute of Zoology for confocal facility and technical support.

## Author contributions

Xin.Y. and B.J. designed the experiments, interpreted the results, and wrote the manuscript. Xin.Y. performed the experiments. H.X. performed bioinformatics analysis. Xu.Y., H.W., H.D., Xiy.Y., Q.Y., F.N.Y., X.Q., and L.Z. provided experimental assistance. Z.H., L.L., and J.T. discussed the project.

## Competing interests

The authors declare no competing interests.
