## [Peer Review File · Nature Communications]

Mcam inhibits macrophage-mediated development of mammary gland through non-canonical Wnt signalingEditorial Note: Parts of this Peer Review File have been redacted as indicated to remove third-party material where no permission to publish could be obtained.

REVIEWER COMMENTS

Reviewer #1 (Remarks to the Author):

This manuscript defines an intriguing role for Mcam in negatively regulating mammary ductal proliferation and extension. The experimental approach is well-structured and systematically dissects and defines a heterotypic intercellular signaling pathway regulating ductal elongation. The authors identified Mcam in CRISP-Cas screens as a negative regulator of in vitro clonogenicity and in vivo mammary regeneration assays. Utilizing an Mcam f/f x MMTV-cre (CKO) mouse they found Mcam loss enhanced macrophage recruitment, ductal elongation, proliferation. Using pharmacological and immunological blockers they demonstrate Mcam CKO phenotype requires IL4/13, stat6 activation, Wnt5a expression in macrophages and Ryk reception in basal cells. These findings are interesting from the perspective of understanding both mammary development and the mechanism of non-canonical Wnt signaling.

Questions and comments concerning the presentation, initial experimental design, and interpretation follow:

1. The authors refer throughout to "stemness". For clarity in the field, it would be better to delete this term throughout the manuscript and instead state the specific biological event they are measuring - clonogenicity, regenerative capacity, proliferation, ductal elongation as appropriate. Extrapolating to the interpretation of "stemness" should be measured and reserved until the discussion. The assays employed (in vitro clonogenicity and in vivo regeneration) measure facultative "stem cell" behavioral characteristics acquired experimentally that are well documented to differ significantly from those of physiological stem cell activity. Similarly, care is needed in the use of the word "MASCs" unless a marker more specific than CD29, which is ubiquitously expressed on all basal cells, has been used to identify them. The authors could bolster their claims by exploring Lgr5, Procr, sSHIP expression in the CKO.
2. Previous studies linking Wnt5a to mammary biology (Roarty and Serra) should be mentioned in the introduction. Omit the discussion on p18 concerning cancer as it does not focus the theme of the paper. It would be interesting to see a discussion of how M-cam may regulate Ryk expression/stability.
3. The use of MMTV-cre to delete Mcam is not ideal, given Mcam expression in basal cells K14-cre would have been more incisive. MMTV is hormonally activated, and expressed chimerically and predominantly in luminal cell types during pregnancy.
4. The data for controls in the transplantation-regeneration assays are of some concern. Success rates for transplantation of 100,000 control WT cells are reported to be much higher in the literature.
5. Controls seem to be missing in Fig. 4L-M?
6. The data show clearly that Mcam loss stimulates ductal proliferation. It is unclear if it additionally promotes differentiation or simply expands of the luminal population expressing such markers. Does it promote ductal differentiation? For example, does progesterone receptor expression remain uniform or become intermittent as in mature ducts?
7. Fig S2 needs to go in the main text and to be discussed in terms of the overall hypothesis of Mcam function during mammary development. The authors argue convincingly that Mcam loss promotes macrophage recruitment during ductal extension, providing a rationale, based on the previous work of Pollard et al, for the observed enhancement of ductal elongation. However, Fig S2F shows that Mcam is also expressed robustly during pregnancy - does Mcam loss affect proliferation /differentiation at this stage? This can easily be assessed by making the transplant recipients pregnant. Importantly, Fig

S2F shows Mcam is maximally expressed during involution – a stage in which macrophage recruitment is instrumental to the restructuring of the gland – what is it doing at this stage?

8. A new and interesting aspect of the paper is the macrophage origin of Wnt5a. The authors mined the scRNAseq dataset of Bach et al (Khaled lab), which is derived from sorted epithelial cell-types, to demonstrate the basal cell expression of Mcam and Ryk mRNAs. Some recent datasets encompass a greater diversity of mammary cell-types - including non-epithelial cells and in some cases focusing on immune cells and macrophages. The overall model presented in Figure 8 would be strengthened if these datasets were investigated and mined to validate Wnt5a expression.

a. Henry et al. Journal of Mammary Gland Biology & Neoplasia vol 26, p43–66 (2021) Characterization of Gene Expression Signatures for the Identification of Cellular Heterogeneity in the Developing Mammary Gland

b. Twigger AJ et al.. Nat Commun. 2022 13(1):562. doi: 10.1038/s41467-021-27895-0. Transcriptional changes in the mammary gland during lactation revealed by single cell sequencing of cells from human milk.

c. Li et al. Cell Rep. 2020 33(13): 108566. doi: 10.1016/j.celrep.2020.108566 Aging-Associated Alterations in Mammary Epithelia and Stroma Revealed by Single-Cell RNA Sequencing

d. Twigger et al Semin Cell Dev Biol. 2021 114: 171–185. Mammary gland development from a single cell 'omics view

e. Azizi et al Cell. 2018 Aug 23; 174(5): 1293–1308.e36. Single-cell Map of Diverse Immune Phenotypes in the Breast Tumor Microenvironment

Reviewer #2 (Remarks to the Author):

What are the noteworthy results?

Hou and colleagues use a genome-scale CRISPR-Cas9 knockout library screening approach to identify gene products that, when knocked out, potentiate the output of colony formation assays in vitro and mammary transplantation assays in vivo. The authors focus further work on the cell adhesion molecule Mcam, which is expressed in CD24+CD29+ basal cells (BC). The authors generate a MMTV-Cre Mcam Floxed mouse and observe elevated frequency of basal cells and, in comparison with WT MECs and BCs, an augmented frequency of mammary repopulation units; however, the frequency of MRUs in WT MECs and BCs observed in this manuscript are lower than those reported in the literature, which complicates the interpretation of the results. Analysis of gene expression profiles in Mcam KO tissues indicates a potential increase in immune cell subsets, which is observed through FACS profiling for macrophages and neutrophils. Conditioned medium taken following the growth of Mcam KO cells increases macrophage migration in vitro implicating a potential recruitment of macrophages in Mcam KO glands but the stringency of the measurements raises questions about the subtle differences that are presented. Macrophages are well known to affect mammary morphogenesis, repopulating ability, and MaSC activity. Through the use of clodronate liposomes to deplete macrophages, the authors demonstrate an inhibitory effect on fat pad filling, which is consistent with published literature (Gyorki et al. 2009). The authors demonstrate heightened IL4 expression in Mcam KO glands and, via anti-IL4 blocking antibodies, describe a requirement for IL4 for macrophage recruitment. Further, the authors demonstrate 400-fold elevated Wnt5a expression in macrophages from Mcam KO glands and, through the use of shRNA, a reliance on Wnt5a for WT or Mcam KO outgrowths. The presented data also indicates the elevated expression of Ryk, and its coexpression with Mcam, in basal cells. Finally, Ryk was needed for the measured colony forming capacity for WT and Mcam KO mammary cells, indicating an effect that is not specific to the Mcam null gland.

In summary, the manuscript uses a discovery screen to identify Mcam as a potential brake on MaSC activity however there are several key controls missing in the description of the screening results.

Through the creation of MMTV-specific Mcam KO mice, and analysis of MaSC activity, they observe subtle changes to (heightened) MaSC activity correlating with elevated macrophage infiltration, and IL4 and Wnt5a expression. The MaSC activity for both WT and Mcam KO basal cells is found to rely on IL4, Wnt5a, and Ryk. For many of the noteworthy findings, however, it is very difficult to critically evaluate the experiments, and interpret the novelty of the findings, because the experimental details are either extremely sparse or lacking in the description of the results and in the Figure legends. For the measurement of MaSC activity, for example, the observed frequencies reported by the authors appear to be 10-fold lower than expected from the literature. The lack of experimental details negatively impacts the consideration of the results, as detailed below.

- Will the work be of significance to the field and related fields and how does it compare to the established literature?

- Does the work support the conclusions and claims, or is additional evidence needed? Are there any flaws in the data analysis, interpretation and conclusions? Do these prohibit publication or require revision?

For many of the noteworthy findings, it is very difficult to critically evaluate the experiments, and interpret the novelty of the findings in relation to the established literature, because the experimental details are either extremely sparse or lacking in the description of the results and in the Figure legends. The lack of details negatively impacts the robustness of the conclusions and claims and makes it extremely difficult to interpret the data, which often shows significant differences between genotypes but differences that range squarely within or below published frequencies.

In Figure 1, Hou and colleagues use a genome-scale CRISPR-Cas9 knockout library screening approach to identify gene products that, when knocked out, potentiate the output of colony formation assays in vitro and mammary transplantation assays in vivo. Sequencing sgRNAs from large clones generated in vitro and in regenerated mammary outgrowths, resulted in a candidate gene list, which was further refined based upon the expression of candidate genes within a published scRNA-sequencing data set. Cumulatively, this analysis implicated Mcam in the regulation of mammary stem cell activity.

The screening approach is similar to that used by Chen et al Cell 2015 (PMID: 25748654), which determined gene products whose loss potentiate metastasis. Unfortunately, the submitted manuscript lacks controls and/or does not provide specifics in the description of the screening approaches and the interpretation of the data (see comment below). Nevertheless, the authors select Mcam for further study as a novel regulator for MaSC activity and show Mcam is expressed in Lin- CD24+CD29+ basal cells. The authors generated MMTV-Cre Mcam floxed mice, confirmed loss of Mcam expression by immunofluorescence and western blot analysis and then identify abnormal epithelial structures, including increases in filled fat pads at 5 weeks (puberty), duct thickness, and the fraction of KI67+ cells.

Comment 1 - It is difficult for this reviewer to comprehend some aspects of the genome-scale screen as it is currently presented in the manuscript. It may be assumed that a sgRNA will be captured through sequencing of large clones if (i) the sgRNA does not alter its targeted gene, (ii) the sgRNA does alter the targeted gene, but that gene is not consequential to clonal growth, or (iii) the sgRNA does alter the targeted gene, and the loss of that gene enables or promotes clonal growth. It is not completely clear from the description of the results or methodologies how the author's distinguished between these possibilities for the sgRNAs that they identified through sequencing.

1.1 How did the researchers confirm the efficacy of the individual sgRNA? For example, did the researchers confirm the efficacy of the Mcam sgRNAs? Did they identify multiple sgRNA targeting Mcam? The data in Figure S2A is presented to support the enrichment of Mcam sgRNA in large clones. But, it is very difficult to interpret this data.

1.2 It appears from the data presented in Fig S1B that several thousands of sgRNA were detected in large clones, for example. Because the researchers state that the MOI would restrict 1 sgRNA to each

cell and, presumably, the clones were generated from single cells. How many clones did the researchers pool prior to sequencing? For example, if the researchers pooled 500 clones (4000 cells per clone, 12 doublings), would you expect 500 sgRNA? The methods simply state that 4 million cells were used for the clone assay, which itself is a bit confusing as the authors describe the clone assay as using 200 MECs.

1.3 It seems that in Fig S1e, 10,000+ detected sgRNA were categorized to give 31 candidates, of which Mcam is one. But, this process of categorization is not well described. It appears in the methods that one basis of the selection was the presence of at least 2 sgRNA clones enriched. It would be helpful to visualize for how many gene products at least 2 sgRNA were enriched in the big clone versus cell library. It would be helpful to see the list of genes with 1, 2, or 3+ sgRNA enriched, and the fraction of samples with those genes enriched, as done in Figure 3 for Chen et al PMID 2574654.

1.4 The screen appears to be designed similar to that described by Chen et al Cell 2015 (PMID: 25748654). But, there are several controls that are presented in Chen et al and are absent from this manuscript. For instance, Chen et al show concordance between technical replicates and biological infection replicates; mention different sgRNAs that target the same gene are correlated in terms of rank change; show candidate genes targeted by two or more sgRNAs isolated in their populations; validate target genes, etc. The current manuscript lacks equivalent analysis of controls and the absence of these controls negatively impacts confidence in the data presented.

Comment 2, on Figure 1 data – The authors propose Mcam KO ducts are thicker in Figure 1I. But, in Figure 1K, the WT duct looks thicker than the Mcam KO duct (and the magnification does not look the same). In Figure 1K, the WT duct looks quite similar to the Mcam KO duct shown in Figure 1I. For the analysis of KI67+ cells in Figure 1K, the Mcam KO duct looks quite similar in structure to the WT TEB in Figure 1I. Because proliferative cells will differ in ducts versus TEB, it is vital to match sections prior to analysis. How did the researchers match sections for their analysis?

Comment 3 – The image quality in Figure 1I and 1K is poor. For the panel of images in Figure 1I, I see only one scale bar, which suggests equal magnifications. But the cell sizes (by nucleus size) are different; the TEB images appear to be a higher magnification. Same issue in Figure 1K where the CKO image appears to be a higher magnification and a smaller duct, which may impact the KI67+ cells.

Comment 4 – The authors are not specific in many of their statements. For example:

4.1. Page 4, at end of introduction : “We found that Mcam mediated the microenvironment of MaSCs as a non-canonical Wnt receptor,...: It is not clear what “mediated the microenvironment” means in relationship to their findings.

4.2. Page 9, A subtle increase in macrophage numbers does not support the conclusion or implication “that macrophages are the predominant mediating cell type in Mcam KO mice”.

4.3. Page 10, the data may support the statement that “depletion of Mcam in the mammary tissue leads to an increase in macrophages”, but it is premature to state “supporting the function of MaSCs”.

4.3 Page 12, “found that systemic ablation of macrophages reversed the increase in MaSC abundance in CKO mice (Figs. 4L-4M),” should read “found that systemic ablation of macrophages reversed the increase in CD24+CD29+ cell abundance in CKO mice (Figs. 4L-4M),”.

In Figure 2, the authors aim to determine the impact of loss of Mcam on MaSC activity, as measured by mammary transplantation assays. First, they provide a FACS examination of Lin-CD24+CD29+ cells, which revealed 3-fold elevation in Mcam KO mice (about 10% of MECs). The authors identify a difference in MRU frequencies in MECs or Lin-CD24+CD29^{high} cells between WT and Mcam KO mice. However, both of these frequencies are substantially lower than published frequencies making it difficult to interpret the significance of these findings.

Comment 5 - The authors performed a limiting dilution MRU assay, but identified very low frequencies (Figure 2D, WT : 1/256,161, range: 1:81,640 (mistake) to 1/80,375; CKO : 1/69,591) compared to reported 1:5000 (Shackleton et al., 2006). But, more surprising is the low frequency of MRU observed

in WT basal cells (1:1023), which is about 10X lower than reported by Gyorki et al 2009 for Lin-CD24+CD29high cells (1:97) or Shackleton et al 2006 (1:64). It is hard to determine the cause of this discrepancy as the Methodology states that a similar protocol was used. Please explain, in particular for the transplantation of CD24+CD29high basal cells, why the MRU frequencies are only 10% of expected.

In Figure 3, the researchers analyze DEGs in Mcam KO versus WT mammary tissues, which indicated potential deregulation of the immune response. Analysis by FACS is not shown but is reported to indicate subtle increases in macrophages (also shown in Figure S4A) and neutrophils. The authors analyse Cx3cr1+ populations by FACS and IF, and these results indicate elevated macrophage levels in Mcam KO tissues although there are concerns with these analyses listed below. But, conditioned medium acquired after culture of CKO MECs augmented the recruitment of macrophages through transwell. Taken together, the data presented in Figure 3 implies a potential recruitment of macrophages in Mcam KO tissues. BUT, the stringency of the protocols and measurements raises questions about the robustness of the subtle differences that are presented.

Comment 6 - It is critical to visualize the FACS profiles and gating strategy used to identify the specified cell types in Figure 3B, even for the cell types that do not show a significant difference between genotypes. Similarly, in Figure 3F, please show the gating strategy. Why do the Cx3cr1 profiles appear to measure events of unspecific size scatter?

Comment 7 – Similar to Figure 1I and Figure 1K, how were the sections chosen in Figure 3E? The Mcam KO “duct” structure (i.e. multiple cell layers) looks quite distinct from that of the WT duct. That is, the sections do not appear to be at equivalent regions.

In Figure 4, the authors examine the importance of macrophages for the observed MaSC activity. Macrophages are well known to affect mammary morphogenesis, repopulating ability, and MaSC activity. Published clonogenic capacity of CD29highCD24+ MECs from wildtype glands is about 15 cells per 100 plated using in vitro colony forming assay (Gyorki et al 2009). So, the observed clonogenic capacity of Mcam KO cells using in vitro mammosphere assay (~13) is greater than the observed for wildtype cells (10) but within the range of published wildtype capacities. Moreover, the inhibitory impact of CL treatment on percent fat pad filling is documented (Gyorki et al 2009), and the results presented in Figure 4H are consistent with this report. However, it is not clearly stated in the Results what tissues were analysed by FACS for Figure 4L-M. The authors indicate CL-treated MECs were implanted in NOD/SCID mice, resulting in no epithelial cells filling the pad (Figure 4K). So, it is not clear what tissue is the source for the CD24+CD29+ epithelial cells shown in the FACS profile in Figure 4L and Figure 4M; could it be CL-treated Mcam KO mice?

Comment 8 – Throughout the description of the results, it is important for the authors to state the specifics of the experiments performed either in the results section or in the Figure legends. For instance, it is not clear how the experiment was performed to give the results presented in Figure 4H. The authors should clearly state the number of CD24+CD29+ cells transplanted into these pads in the Results (I believe it states 200 cells in the methods). Similarly, for Figure 4K, it is not clear how many MECs were injected; judging from Figure 2D it could be 10,000 or 100,000. Finally, for Figure 4L and 4M, it is not clear what is the source of the epithelium as Figure 4K indicates that the treatment with CL prevents fat pad filling.

In Figure 5, the authors demonstrate heightened IL4 expression, which correlates with Stat6 phosphorylation and mediates macrophage migration in vitro and in vivo. Again, however, the experimental details are absent for the analysis of in vivo macrophage infiltration leaving the reader to assume that Figure 5F – 5K are mammary repopulating assays. But, these experiments may be examining intact WT or Mcam KO glands treated with anti-IL4 and, thus, confirm the effect of anti-IL4 on macrophage recruitment (in Figure 5F – 5I). Again, the specific experimental methodology is not clear.

Canonical Wnt signaling pathways did not seem to discriminate between WT and CKO glands but the expression of Wnt5a was dramatically elevated at both the transcript expression and protein expression levels. Next, various cell populations were isolated (but, representative FACS profiles were not provided) and Wnt5a expression was shown to be produced by F4/80+ macrophages isolated in stroma. Macrophages were isolated and treated with a non-hairpin shRNA control or a shRNA targeting Wnt5a and transplanted with CD24+CD29+ cells isolated from either of WT or CKO glands. The efficacy of Wnt5a knockdown is not shown. But, the intervention equivalently blocked the outgrowths from WT or CKO basal cells, indicating an effect that is not specific to the Mcam null gland. Similarly, in Figure 7, the author's present data that indicates the elevated expression of Ryk, and its coexpression with Mcam, in basal cells. Anti-Ryk antibody treatment reduced colony forming capacity for WT and CKO mammary cells, indicating an effect that is not specific to the Mcam null gland.

- Is the methodology sound? Does the work meet the expected standards in your field?**
- Is there enough detail provided in the methods for the work to be reproduced?**

In summary, it is very difficult to critically evaluate the experiments, and interpret the novelty of the findings, because the methodological details are either extremely sparse or completely lacking in the description of the results and in the Figure legends. The presentation of the data generated in the initial sgRNA screen is lacking multiple key controls and the specific process that identified Mcam as a candidate is not clear. As well, the image quality of tissue sections is poor in general and the sections do not seem to be appropriately matched between genotypes. Finally, representative FACS profiles and gating strategies should be included.

Reviewer #1 (Remarks to the Author):

This manuscript defines an intriguing role for Mcam in negatively regulating mammary ductal proliferation and extension. The experimental approach is well-structured and systematically dissects and defines a heterotypic intercellular signaling pathway regulating ductal elongation. The authors identified Mcam in CRISP-Cas screens as a negative regulator of in vitro clonogenicity and in vivo mammary regeneration assays. Utilizing an Mcam f/f x MMTV-cre (CKO) mouse they found Mcam loss enhanced macrophage recruitment, ductal elongation, proliferation. Using pharmacological and immunological blockers they demonstrate Mcam CKO phenotype requires IL4/13, stat6 activation, Wnt5a expression in macrophages and Ryk reception in basal cells. These findings are interesting from the perspective of understanding both mammary development and the mechanism of non-canonical Wnt signaling.

Questions and comments concerning the presentation, initial experimental design, and interpretation follow:

1. The authors refer throughout to “stemness”. For clarity in the field, it would be better to delete this term throughout the manuscript and instead state the specific biological event they are measuring - clonogenicity, regenerative capacity, proliferation, ductal elongation as appropriate. Extrapolating to the interpretation of “stemness” should be measured and reserved until the discussion. The assays employed (in vitro clonogenicity and in vivo regeneration) measure facultative “stem cell” behavioral characteristics acquired experimentally that are well documented to differ significantly from those of physiological stem cell activity. Similarly, care is needed in the use of the word “MASCs” unless a marker more specific than CD29, which is ubiquitously expressed on all basal cells, has been used to identify them. The authors could bolster their claims by exploring Lgr5, Procr, sSHIP expression in the CKO.

Response: Thank you for your suggestions to improve the manuscript. Regarding MaSCs, we re-checked the whole text and replaced them with mammary epithelial

cells (MECs) and regenerative capacity of MECs for more accurate expression. In addition, we discussed the potential effects by examining the expression of stemness-related genes, including *K14*, *Lgr5*, *Procr*, and *sSHIP*, in basal cells derived from the mammary tissues of WT and CKO mice by MMTV-Cre. These genes were highly elevated in CKO mice compared with WT mice (Fig. S5C), supporting the potential regulatory roles of *Mcam* for MaSCs.

Figure S5C. qRT-PCR results for *K14*, *Procr*, *Lgr5*, and *sSHIP* in basal cells isolated from mammary tissues of *Mcam* WT or CKO mice. n = 3 replicates in each group.

2. Previous studies linking *Wnt5a* to mammary biology (Roarty and Serra) should be mentioned in the introduction. Omit the discussion on *p18* concerning cancer as it does not focus the theme of the paper. It would be interesting to see a discussion of how *M-cam* may regulate *Ryk* expression/stability.

Response: Thank you for your suggestion. We have cited the previous studies about the roles of *Wnt5a* in mammary biology (2nd paragraph on page 4). The discussion on cancer has been deleted. In addition, discussion on the potential mechanisms regarding how *Mcam* regulates *Ryk* expression/stability has been added to the 1st paragraph of the Discussion section.

3. The use of MMTV-cre to delete *Mcam* is not ideal, given *Mcam* expression in basal cells *K14-cre* would have been more incisive. MMTV is hormonally activated, and

expressed chimerically and predominantly in luminal cell types during pregnancy.

Response: Thank you for your suggestion.

The primary objective of this study was to investigate the effects of Mcam on the mammary gland at the virgin stage. The MMTV promoter is well-recognized for its early activation at less than 1 week of age. Thus, we applied the MMTV-Cre line to examine the pubertal effects of Mcam. Moreover, although Mcam is predominantly expressed in the basal layer of the mammary gland, there is also little expression in the luminal layer. Therefore, we chose MMTV-Cre mice to deplete Mcam with high penetration.

We agree that K14-Cre mouse strain is a great model for confirming the phenotype and mechanism we observed. We therefore depleted Mcam using K14-Cre mice in the revised manuscript. As the experimental approach using MMTV-Cre mice, Mcam flox mice were bred with K14-Cre mice to generate Mcam KO mice with specific loss of Mcam in basal cells. After effectively knocking out Mcam using K14-Cre mice (Figs. S3C-S3E), we found that loss of Mcam significantly lengthened the mammary ducts in 5-week-old mice, and therefore increased mammary ductal occupation in the fat pads (Figs. S3F-S3G). In addition, mammary ductal thickness increased significantly (Figs. S3I-S3J). IF staining with Ki67 (cell proliferation marker) showed that both basal and luminal cells had much higher percentages of Ki67-positive cells upon loss of Mcam (Figs. S4D-S4E). Similarly, to investigate whether the above phenotypes were driven by MaSCs, we explored changes in the MaSC-enriched cell population ($\text{Lin}^- \text{CD24}^+ \text{CD29}^+$) in heterozygous CKO mice ($\text{Mcam}^{\text{fl/wt}}/\text{K14-Cre}$). Compared to $\text{Mcam}^{\text{fl/wt}}$ (littermate control) mice, the CKO mice exhibited a much higher basal population (Figs. S5A-S5B), indicating that Mcam loss induces the basal cell compartment containing MaSCs. Clone assay showed Mcam depletion significantly increased the numbers and diameters of colonies (S5E-S5G), representing MECs clonogenicity and proliferation ability. Mechanistically, we examined changes in macrophages and found both CD206^+ and Cx3cr1^+ macrophages increased significantly in CKO mice (Fig. S7D). To determine why macrophages increased, we examined changes in the Il4-Stat6 axis and found that both Il4 and

p-stat6 levels increased in CKO mice (Figs. S9A-S9B). IHC staining showed that the expression levels of Wnt5a and Ryk were increased in CKO mice (Figs. S9D and S10B). The above phenotypes were consistent with those in CKO mice driven by MMTV-Cre, further highlighting the role of *Mcam* in mammary gland development.

Figures S3–S5, S7, and S9–S10. (S3C) qRT-PCR analysis of *Mcam* mRNA expression in MECs from *Mcam*^{fl/wt} and *Mcam*^{fl/wt/K14-Cre} mice. $n = 5$ mice per genotype. (S3D) IF staining showing KO efficiency of *Mcam*. Scale bar, 50 μ m.

(S3E) Western blotting showing Mcam protein expression in Mcam^{fl/wt} and Mcam^{fl/wt}/K14-Cre MECs. Three independent mice for each group. (S3F-S3G) Representative images (S3F) and their quantification (S3G) of mammary duct extension of whole-mount staining in Mcam^{fl/wt} and Mcam^{fl/wt}/K14-Cre mice at 5 weeks of age. (S3I) Representative images of mammary tissues in 5-week-old Mcam^{fl/wt} and Mcam^{fl/wt}/K14-Cre mice by H&E staining. Scale bar, 20 μ m. (S3J) Quantification of mammary duct thickness in Mcam^{fl/wt} and Mcam^{fl/wt}/K14-Cre mice. n = 22 ducts in Mcam^{fl/wt} mice, n = 18 ducts in Mcam^{fl/wt}/K14-Cre. (S4D) Immunostaining of α -SMA (red) and Ki67 (green) expression in Mcam^{fl/wt} and Mcam^{fl/wt}/K14-Cre mice. Scale bar, 50 μ m. (S4E) Ki67⁺ cells in basal and luminal cell populations of Mcam^{fl/wt} and Mcam^{fl/wt}/K14-Cre mice. n = 11 sections in each group. (S5A-S5B) FACS analysis (S5A) and statistical ratios of basal cell population (Lin⁻CD24⁺CD29⁺) (S5B) in Mcam^{fl/wt} and Mcam^{fl/wt}/K14-Cre mice. Five individual mice for each genotype. (S5E-S5G) Representative images (E) and statistical results of clone numbers (F, n = 18), clone diameters greater than 100 μ m (G, n = 40) of colonies formed by WT and CKO (depleted by K14-Cre mice) MECs, respectively. Scale bar, 100 μ m. (S7D) IF images of CD206⁺ macrophages and Cx3cr1⁺ macrophages in Mcam^{fl/wt} and Mcam^{fl/wt}/K14-Cre mice. Scale bar, 50 μ m for CD206 staining, and 20 μ m for Cx3cr1 staining. (S9A) qRT-PCR analysis of expression levels of Il4 in MECs from Mcam^{fl/wt} and Mcam^{fl/wt}/K14-Cre mice. n = 3 replicates in each group. (S9B) Western blotting showing Stat6 and p-Stat6 levels in Mcam^{fl/wt} and Mcam^{fl/wt}/K14-Cre mammary tissues. Three independent mice in each group. (S9D) IHC staining of Wnt5a in Mcam^{fl/wt} and Mcam^{fl/wt}/K14-Cre mice. Scale bar, 50 μ m. (S10B) IHC staining of Wnt5a in Mcam^{fl/wt} and Mcam^{fl/wt}/K14-Cre mice. Scale bar, 50 μ m.

4. The data for controls in the transplantation-regeneration assays are of some concern. Success rates for transplantation of 100,000 control WT cells are reported to be much higher in the literature.

Response: Thank you for your critical reading.

For this transplantation assay, the filled percentages of mammary ducts in the fat pads were counted to calculate success rates. Notably, the earlier the dissection time points of the fat pad, the lower the frequencies of successful duct regeneration. In previous reports, regenerated mammary tissues were harvested during the mature stage (8–12 weeks (Cai et al., 2017) and 6–8 weeks (Chakrabarti et al., 2014)), while we selected the premature stage (5 weeks) for our experiments to capture differences between the Mcam WT and CKO groups. As such, the regeneration frequencies of 100,000 control WT cells were lower than those reported in the literature. We selected the 5-week-stage as Mcam acts as an inhibitor to delay ductal elongation. At the mature stages, both the WT and CKO groups reached the maximum limits of the fat pad, leading to indistinguishable filled percentages. Earlier reports on mammary duct inhibitors (Bernardo et al., 2010; Chakrabarti et al., 2014; Shackleton et al., 2006) have also selected the premature stage for assessing endpoints. To confirm this, we repeated the transplantation assay, harvesting the regenerated mammary tissues at a later stage (6 weeks). As shown in Fig. 2E, a higher success rate was achieved, thereby validating our experimental procedures.

Figure 2E. Representative images (left) and reconstitution efficiency at limiting dilution (right) of whole mount-stained mammary outgrowths derived from transplantation of WT or CKO Lin⁻ MEC cells. Scale bar, 1 mm. n = 8 for each group.

5. Controls seem to be missing in Fig. 4L-M?

Response: Thank you. We have added the control data in Figs. 4L-4M in the

revised manuscript.

Figure 4. (4L-4M) FACS analysis (4L) and quantification (4M) of basal population in WT and CKO mice treated with or without CL. n = 5 mice in each group.

6. The data show clearly that Mcam loss stimulates ductal proliferation. It is unclear if it additionally promotes differentiation or simply expands of the luminal population expressing such markers. Does it promote ductal differentiation? For example, does progesterone receptor expression remain uniform or become intermittent as in mature ducts?

Response: Thank you for your suggestion. In addition to promoting luminal population proliferation (Figs. 2A-2B in revised manuscript), Mcam loss also promoted differentiation (Fig. S4G in revised manuscript). We examined the expression of luminal differentiation markers, including *Stat5*, *Elf5*, *Foxa1*, *Esr1*, and *Gata3*, which were highly up-regulated in CKO mice.

According to your suggestion, we also examined the expression of the progesterone receptor (PR) and found that both PR⁻ and PR⁺ cells were present in CKO mice (Fig. S4H), suggesting PR expression becomes intermittent in mature ducts.

Figure S4H. IHC staining of PR in WT and CKO mammary tissues. Scale bar, 20 μ m.

7. Fig S2 needs to go in the main text and to be discussed in terms of the overall hypothesis of Mcam function during mammary development. The authors argue convincingly that Mcam loss promotes macrophage recruitment during ductal extension, providing a rationale, based on the previous work of Pollard et al, for the observed enhancement of ductal elongation. However, Fig S2F shows that Mcam is also expressed robustly during pregnancy – does Mcam loss affect proliferation /differentiation at this stage? This can easily be assessed by making the transplant recipients pregnant. Importantly, Fig S2F shows Mcam is maximally expressed during involution – a stage in which macrophage recruitment is instrumental to the restructuring of the gland – what is it doing at this stage?

Response: Thank you for your questions and suggestions. As suggested, we moved the original Figs. S2D and S2F to the main Figs. 1E and 1F (described on Page 7 in the revised manuscript).

Our main findings indicated that Mcam suppressed the clonogenicity and regenerative capacity of the mammary gland. To delve deeper into this phenomenon, we specifically focused on the pubertal stage because the other stages, such as pregnancy and involution, are intricately regulated by hormones and pose greater complexity for studying regenerative capacity.

Exploring the expression of Mcam during the pregnant stage raises intriguing questions regarding its roles throughout the various developmental stages. As suggested, we assessed this phenotype in pregnant WT and CKO mice and evaluated

proliferation and differentiation using PR and Ki67 antibodies as markers. IHC staining showed no significant difference between WT and CKO mammary tissues (Panel A in response letter), suggesting that Mcam loss does not affect proliferation/differentiation at the pregnancy stage. This may be because Mcam does not affect clonogenicity and regenerative capacity at the pregnancy stage, aligning with previous reports showing unchanged mammary stem cells (Fu et al., 2020), thereby supporting our major statement.

To gain a comprehensive understanding of Mcam expression during involution, we collected samples spanning various stages, ranging from 6 h to 6 days after weaning, including reversible and irreversible involution. The staining of WT and CKO mammary tissues showed no Mcam signals at any involution time point (Fig. 1E). As mentioned, the absence of Mcam is consistent with the robust recruitment of macrophages, a key process for mammary gland restructuring during involution (O'Brien et al., 2010). We acknowledge your comment regarding the images in the original figures (involution in original Fig. S2F). The solid red signals in those images, indicative of nuclear localization, do not correspond to Mcam expression. Mcam is widely recognized as a membrane protein primarily expressed in basal cells (Wang et al., 2020). We apologize for any confusion caused by providing misleading images.

Figure 1. (Panel A) IHC staining of PR and Ki67 in WT and CKO mammary tissues at pregnant stage. Scale bar, 50 μ m. (1E) Immunostaining of Mcam expression in WT and CKO mice. Scale bar, 50 μ m. I, involution. n = 3 mice at each time point.

8. A new and interesting aspect of the paper is the macrophage origin of Wnt5a. The

authors mined the scRNAseq dataset of Bach et al (Khaled lab), which is derived from sorted epithelial cell-types, to demonstrate the basal cell expression of Mcam and Ryk mRNAs. Some recent datasets encompass a greater diversity of mammary cell-types - including non-epithelial cells and in some cases focusing on immune cells and macrophages. The overall model presented in Figure 8 would be strengthened if these datasets were investigated and mined to validate Wnt5a expression.

a. Henry et al. Journal of Mammary Gland Biology & Neoplasia vol 26, p43–66 (2021) Characterization of Gene Expression Signatures for the Identification of Cellular Heterogeneity in the Developing Mammary Gland

b. Twigger AJ et al., Nat Commun. 2022 13(1):562. doi: 10.1038/s41467-021-27895-0. Transcriptional changes in the mammary gland during lactation revealed by single cell sequencing of cells from human milk.

c. Li et al. Cell Rep. 2020 33(13): 108566. doi: 10.1016/j.celrep.2020.108566 Aging-Associated Alterations in Mammary Epithelia and Stroma Revealed by Single-Cell RNA Sequencing

d. Twigger et al Semin Cell Dev Biol. 2021 114: 171–185. Mammary gland development from a single cell ‘omics view

e. Azizi et al Cell. 2018 Aug 23; 174(5): 1293–1308.e36. Single-cell Map of Diverse Immune Phenotypes in the Breast Tumor Microenvironment

Response: Thank you for your helpful suggestion to strengthen the Wnt5a expression levels in macrophages.

We have revalidated the expression pattern of Wnt5a in non-epithelial cells using suggested scRNA-seq data from mammary gland. The analysis results demonstrated that three datasets showed WNT5A was highly expressed in macrophages (Panel B (Wu et al., 2020), Panel C (Twigger et al., 2022), Panel D (Azizi et al., 2018)). One dataset (Henry et al., 2021) was not downloadable, thus it was not used for further analysis. In addition, one dataset shows that Wnt5a was nearly undetectable in all immune cells (Panel E (Li et al., 2020)). We thought this might be due to the lower captured RNAs in immune cells compared with non-immune cells in mammary gland, thus there were less detectable UMIs (unique molecular identifiers) and genes in

mouse immune cells (Panel E). In general, the published sc-RNA-seq data in intact mammary tissues support the high Wnt5a expression levels in macrophages.

Figure (Panel B–E) The detection of UMIs and genes for each cell population in each sample and the expression pattern of Wnt5a in human (Panel B-Panel D) and mouse (Panel E) mammary tissues.

Reviewer #2 (Remarks to the Author):

What are the noteworthy results?

Hou and colleagues use a genome-scale CRISPR-Cas9 knockout library screening approach to identify gene products that, when knocked out, potentiate the output of colony formation assays in vitro and mammary transplantation assays in vivo. The

authors focus further work on the cell adhesion molecule Mcam, which is expressed in CD24+CD29+ basal cells (BC). The authors generate a MMTV-Cre Mcam Floxed mouse and observe elevated frequency of basal cells and, in comparison with WT MECs and BCs, an augmented frequency of mammary repopulation units; however, the frequency of MRUs in WT MECs and BCs observed in this manuscript are lower than those reported in the literature, which complicates the interpretation of the results. Analysis of gene expression profiles in Mcam KO tissues indicates a potential increase in immune cell subsets, which is observed through FACS profiling for macrophages and neutrophils. Conditioned medium taken following the growth of Mcam KO cells increases macrophage migration in vitro implicating a potential recruitment of macrophages in Mcam KO glands but the stringency of the measurements raises questions about the subtle differences that are presented. Macrophages are well known to affect mammary morphogenesis, repopulating ability, and MaSC activity. Through the use of clodronate liposomes to deplete macrophages, the authors demonstrate an inhibitory effect on fat pad filling, which is consistent with published literature (Gyorki et al. 2009). The authors demonstrate heightened IL4 expression in Mcam KO glands and, via anti-IL4 blocking antibodies, describe a requirement for IL4 for macrophage recruitment. Further, the authors demonstrate 400-fold elevated Wnt5a expression in macrophages from Mcam KO glands and, through the use of shRNA, a reliance on Wnt5a for WT or Mcam KO outgrowths. The presented data also indicates the elevated expression of Ryk, and its coexpression with Mcam, in basal cells. Finally, Ryk was needed for the measured colony forming capacity for WT and Mcam KO mammary cells, indicating an effect that is not specific to the Mcam null gland.

In summary, the manuscript uses a discovery screen to identify Mcam as a potential brake on MaSC activity however there are several key controls missing in the description of the screening results. Through the creation of MMTV-specific Mcam KO mice, and analysis of MaSC activity, they observe subtle changes to (heightened) MaSC activity correlating with elevated macrophage infiltration, and IL4 and Wnt5a expression. The MaSC activity for both WT and Mcam KO basal cells

is found to rely on IL4, Wnt5a, and Ryk. For many of the noteworthy findings, however, it is very difficult to critically evaluate the experiments, and interpret the novelty of the findings, because the experimental details are either extremely sparse or lacking in the description of the results and in the Figure legends. For the measurement of MaSC activity, for example, the observed frequencies reported by the authors appear to be 10-fold lower than expected from the literature. The lack of experimental details negatively impacts the consideration of the results, as detailed below.

Response: Thank you very much for your careful reading and valuable comments. For the concerns mentioned (e.g., frequency of MRUs, stringency of measurements, etc.), we have provided new supporting data or experimental details, as shown in the following comments.

- Will the work be of significance to the field and related fields and how does it compare to the established literature?

- Does the work support the conclusions and claims, or is additional evidence needed? Are there any flaws in the data analysis, interpretation and conclusions?

Do these prohibit publication or require revision?

For many of the noteworthy findings, it is very difficult to critically evaluate the experiments, and interpret the novelty of the findings in relation to the established literature, because the experimental details are either extremely sparse or lacking in the description of the results and in the Figure legends. The lack of details negatively impacts the robustness of the conclusions and claims and makes it extremely difficult to interpret the data, which often shows significant differences between genotypes but differences that range squarely within or below published frequencies.

In Figure 1, Hou and colleagues use a genome-scale CRISPR-Cas9 knockout library screening approach to identify gene products that, when knocked out, potentiate the output of colony formation assays in vitro and mammary transplantation assays in vivo. Sequencing sgRNAs from large clones generated in vitro and in regenerated mammary outgrowths, resulted in a candidate gene list, which was further refined

based upon the expression of candidate genes within a published scRNA-sequencing data set. Cumulatively, this analysis implicated Mcam in the regulation of mammary stem cell activity.

The screening approach is similar to that used by Chen et al Cell 2015 (PMID: 25748654), which determined gene products whose loss potentiate metastasis. Unfortunately, the submitted manuscript lacks controls and/or does not provide specifics in the description of the screening approaches and the interpretation of the data (see comment below). Nevertheless, the authors select Mcam for further study as a novel regulator for MaSC activity and show Mcam is expressed in Lin-CD24+CD29+ basal cells. The authors generated MMTV-Cre Mcam floxed mice, confirmed loss of Mcam expression by immunofluorescence and western blot analysis and then identify abnormal epithelial structures, including increases in filled fat pads at 5 weeks (puberty), duct thickness, and the fraction of KI67+ cells.

Comment 1 - It is difficult for this reviewer to comprehend some aspects of the genome-scale screen as it is currently presented in the manuscript. It may be assumed that a sgRNA will be captured through sequencing of large clones if (i) the sgRNA does not alter its targeted gene, (ii) the sgRNA does alter the targeted gene, but that gene is not consequential to clonal growth, or (iii) the sgRNA does alter the targeted gene, and the loss of that gene enables or promotes clonal growth. It is not completely clear from the description of the results or methodologies how the author's distinguished between these possibilities for the sgRNAs that they identified through sequencing.

Response: Thank you for your excellent suggestions. To address the rationale behind selecting large clones, we added description regarding the screening and identification of sgRNAs enriched in large clones in the Methods section (1st paragraph on page 25).

In our experimental procedure, a large quantity of sgRNAs was transduced into the mammary cells, resulting in their enrichment through the formation of colonies. As stated, the following criteria were used for selecting sgRNAs: (i) If the sgRNA failed to induce any alterations in its targeted gene and colonies either did not form or

remained small in size, then it was not considered for further analysis. (ii) If the sgRNA induced alterations in the targeted gene but the gene did not significantly impact clonal growth, it was not selected as it shared the same characteristics as described in section (i). (iii) If the sgRNA induced alterations in the targeted gene, and the loss of that gene promoted or enabled clonal growth, it was considered enriched and was subjected to deep sequencing.

1.1 How did the researchers confirm the efficacy of the individual sgRNA? For example, did the researchers confirm the efficacy of the Mcam sgRNAs? Did they identify multiple sgRNA targeting Mcam? The data in Figure S2A is presented to support the enrichment of Mcam sgRNA in large clones. But, it is very difficult to interpret this data.

Response: Thank you for your comments.

The GeCKOv2 library was developed in a previous report, which demonstrated high KO efficacy of individual sgRNAs (Shalem et al., 2014). This library has been applied by the other groups independently, proving its efficacy for various target genes (Ruiz et al., 2016). After purchasing commercial GeCKOv2 library plasmid from the Addgene Company (#1000000052), we randomly selected several sgRNAs to validate their efficacy. All tested sgRNA sequences were effectively depleted of target genes (Panel F below). To validate the efficacy of multiple sgRNAs targeting *Mcam* from the GeCKOv2 library, we infected primary mammary epithelial cells with sgRNAs of *Mcam* and negative control lentivirus. Results showed that all *Mcam* sgRNAs were effectively knocked out (Panel G below). These findings provide compelling evidence regarding the high quality and effectiveness of the sgRNAs contained within the library.

The original Fig. S2A showed the fold-changes in each *Mcam* sgRNA between various sample pairs, i.e., enrichment of *Mcam* sgRNA in each sample. The red lines represent enriched individual sgRNAs between various comparisons, and the blue ones represent non-enriched sgRNAs. This description was added in the revised legend of S2C.

Figure (Panel F–G) Efficacy of individual sgRNAs targeting FoxM1 (Panel F) and Mcam (Panel G), as validated by western blotting.

1.2 It appears from the data presented in Fig S1B that several thousands of sgRNA were detected in large clones, for example. Because the researchers state that the MOI would restrict 1 sgRNA to each cell and, presumably, the clones were generated from single cells. How many clones did the researchers pool prior to sequencing? For example, if the researchers pooled 500 clones (4000 cells per clone, 12 doublings), would you expect 500 sgRNA? The methods simply state that 4 million cells were used for the clone assay, which itself is a bit confusing as the authors describe the clone assay as using 200 MECs.

Response: Thank you for your comments. To enrich the sgRNAs that promote clonogenicity of mammary stem cells, 4×10^6 cells (nearly $40 \times$ coverage) were applied for clonal assay. After 10–14 days, all large clones (diameter $> 100 \mu\text{m}$) were collected by manually picking for sequencing. The number of large clones was in the thousands, consistent with the number of sgRNAs detected in large clones after sequencing.

1.3 It seems that in Fig S1e, 10,000+ detected sgRNA were categorized to give 31 candidates, of which Mcam is one. But, this process of categorization is not well described. It appears in the methods that one basis of the selection was the presence of at least 2 sgRNA clones enriched. It would be helpful to visualize for how many gene products at least 2 sgRNA were enriched in the big clone versus cell library. It would be helpful to see the list of genes with 1, 2, or 3+ sgRNA enriched, and the

fraction of samples with those genes enriched, as done in Figure 3 for Chen et al PMID 2574654.

Response: Thank you for your comments and suggestions. In our study, we hypothesized that if a gene played a negative regulatory role in mammary ductal proliferation and extension, the corresponding sgRNAs should be enriched in large clones or regenerated mammary tissues obtained from transplantation assays, in comparison to the sgRNA cell library and small clones.

We agree with your previous comment that “It may be assumed that a sgRNA will be captured through sequencing of large clones if (i) the sgRNA does not alter its targeted gene, (ii) the sgRNA does alter the targeted gene, but that gene is not consequential to clonal growth, or (iii) the sgRNA does alter the targeted gene, and the loss of that gene enables or promotes clonal growth”. Furthermore, we considered that if a sgRNA altered its targeted gene and loss of that gene enabled clonal growth, then multiple sgRNAs targeting this clonal growth-promoting gene would more likely be enriched in the corresponding samples. Additionally, we considered that if a sgRNA altered its targeted gene and loss of that gene enabled clonal growth, the target gene would be highly expressed in basal cells, as such cells participate in clonal growth, and loss of gene expression in basal cells would not promote clonal growth. Taking these factors into consideration, we selected candidate sgRNAs with possible biological effects on clonal growth based on the following conditions: (1) the sgRNA-targeted gene had at least two enriched sgRNAs in the large clone samples compared to the cell library and small clone samples, (2) the sgRNA-targeted gene had at least two enriched sgRNAs in the mammary tissues generated from transplantation assays compared to the sgRNA cell library, and (3) the sgRNA-targeted gene was more highly expressed in basal cells than in luminal cells. Genes for which target sgRNAs met the above conditions were regarded as potential candidates responsible for clonal growth. We have revised the analysis methods in the new manuscript accordingly.

Per your suggestion, we compiled a list of genes (Supplementary Table S1) showing enrichment of 1, 2, or 3 or more sgRNAs, along with the proportion of

samples where these genes demonstrated enrichment, as illustrated in Fig. 3 of Chen et al. (2015). We also created corresponding figures, which shown in Fig. S1E.

Figure S1E. Number of genes with 1, 2, or 3+ significantly enriched sgRNAs targeting that gene in each sample.

1.4 The screen appears to be designed similar to that described by Chen et al Cell 2015 (PMID: 25748654). But, there are several controls that are presented in Chen et al and are absent from this manuscript. For instance, Chen et al show concordance between technical replicates and biological infection replicates; mention different sgRNAs that target the same gene are correlated in terms of rank change; show candidate genes targeted by two or more sgRNAs isolated in their populations; validate target genes, etc. The current manuscript lacks equivalent analysis of controls and the absence of these controls negatively impacts confidence in the data presented.

Response: Thank you for your comments and questions. The work of Chen et al. (2015) aimed to demonstrate that Cas9-based screening was a robust method to systematically assay gene phenotypes in cancer evolution *in vivo*. Hence, they provided detailed systematic measurements and showed the performance of the Cas9 screening library. In contrast, we aimed to screen candidate genes that may account

for clonal growth and investigated the molecular mechanism of specific candidate genes but did not focus on verifying library effectiveness given its wide use by many independent research groups. Thus, our aims differed from those of Chen et al. (2015).

In our manuscript, we used biological infection replicates for the transplantation assays, such as S5971–S5980. Concordance between samples is shown in Fig. S1A and candidate genes targeted by two or more sgRNAs are shown in Figs. S2A-S2B in the revised manuscript. Following our goal, we selected one candidate gene, *Mcam*, to validate the biological function and molecular mechanisms, rather than validate all target genes. According to your comments, we provided several controls in the manuscript, as presented in Figs. S1-S2.

Figure S2. (S2A) Overlap analysis of top enriched sgRNAs in large clones and regenerated mammary tissues, in which target genes were more highly expressed in basal cells than luminal cells. (S2B) Candidate genes targeted by two or more sgRNAs in large clones and regenerated mammary tissues.

Comment 2, on Figure 1 data – The authors propose Mcam KO ducts are thicker in Figure 1I. But, in Figure 1K, the WT duct looks thicker than the Mcam KO duct (and the magnification does not look the same). In Figure 1K, the WT duct looks quite similar to the Mcam KO duct shown in Figure 1I. For the analysis of KI67+ cells in Figure 1K, the Mcam KO duct looks quite similar in structure to the WT TEB in Figure 1I. Because proliferative cells will differ in ducts versus TEB, it is vital to match sections prior to analysis. How did the researchers match sections for their analysis?

Response: Thank you for your comments and questions.

Regarding the thickness of ductal cell layers depicted in the original Figs. 1I and 1K, we specifically counted the layers of epithelial cells, which are densely packed and contribute to the formation of the ductal cavity. The stromal cell layer, which is looser due to the more appearance of extracellular matrix, was excluded from the measurements. This exclusion was based on the use of the MMTV-Cre mouse line, which specifically activates Cre recombinase in epithelial cells but not in other cell types (Zeng et al., 2020). The distinction of the mammary epithelial cell layer in H&E staining images was made based on the density of the cell layer (Fig. S3H). Furthermore, the outer layer of epithelial cells, i.e., basal cells, was visualized through staining with the cell type-specific marker K14 (cytokeratin 14) (Fig. 2A). Therefore, in our measurements, we could count the epithelial cell layer specifically. In the original Fig. 1K, only K14-labeled cells and their inner layer were measured (see inset in Fig. S4A).

Based on these principles, it explains why the WT duct appeared thicker than the Mcam KO duct in the original Fig. 1K, while appearing quite similar to the Mcam KO duct in the original Fig. 1I. For analysis, we measured the thickness of more than 30 ducts using this criterion, as shown in Fig. 1L. The K14-staining results further demonstrated the thicker epithelial cell layer in CKO samples than in WT samples (Figs. 2A-2B, S4B). We apologize for the lack of detailed information in the original manuscript, which may have caused misunderstandings and for the potential confusion in the original Fig. 1K. As a result, we selected new representative pictures

from the images below to present this data in a clearer manner.

Thank you also for your question regarding TEB structure. As there is no well-recognized marker to distinguish TEB from ducts, they can only be defined based on their morphological characteristics. TEBs are usually characterized by a closed or near closed and rounded luminal structure (Hovey et al., 2011). To clarify this, we recounted the TEB structures, focusing on those with a closed and rounded structure, as shown in Fig. S4C. Furthermore, we have recalculated the proportion of proliferating cells within each population of ducts and TEB structures, respectively (Fig. 2B).

Figures S3 and S4. (S3H) Schematic of measurement methods in H&E staining. Scale bar, 20 μ m. Red symbol indicates measured thickness. (S4A) Schematic of measurement methods in IF staining. Scale bar, 50 μ m. White

symbol indicates measured thickness. (S4B) Immunostaining of K14 (red) and Ki67 (green) expression in duct structure of WT and CKO mice. Scale bar, 20 μm . (S4C) Immunostaining of K14 (red) and Ki67 (green) expression in TEB structure of WT and CKO mice. Scale bar, 20 μm . (2B) Ki67⁺ cells in basal and luminal cell populations of duct and TEB structure in WT and CKO mice. n = 10 sections in each group.

Comment 3 – The image quality in Figure 1I and 1K is poor. For the panel of images in Figure 1I, I see only one scale bar, which suggests equal magnifications. But the cell sizes (by nucleus size) are different; the TEB images appear to be a higher magnification. Same issue in Figure 1K where the CKO image appears to be a higher magnification and a smaller duct, which may impact the KI67+ cells.

Response: Thank you for your critical comments. We replaced the low-resolution pictures in Fig. 1K and recalculated the thicknesses of the ducts and TEBs in WT and CKO mice (Fig. 1L). We apologize for the missing scale bars, which have been added to demonstrate corresponding magnification in both the TEB and ductal pictures (Figs. 2A-2B).

Figures 1 and 2. (1K) Representative images of mammary tissues in

5-week-old WT and CKO mice by H&E staining. Scale bar, 20 μ m. (1L) Quantification of mammary duct and TEB thickness in WT and CKO mice. n = 50 ducts or n = 30 TEBs in WT and CKO mice. (2A) Immunostaining of K14 (red) and Ki67 (green) expression levels in ducts and TEB structures of WT and CKO mice. Scale bar, 20 μ m. (2B) Ki67⁺ cells in basal and luminal cell populations of ducts and TEB structures in WT and CKO mice. n = 10 sections in each group.

Comment 4 – The authors are not specific in many of their statements. For example:

4.1. Page 4, at end of introduction: “We found that Mcam mediated the microenvironment of MaSCs as a non-canonical Wnt receptor,...: It is not clear what “mediated the microenvironment” means in relationship to their findings.

Response: Thank you for your comments. This has been corrected.

4.2. Page 9, A subtle increase in macrophage numbers does not support the conclusion or implication “that macrophages are the predominant mediating cell type in Mcam KO mice”.

Response: Thank you for your comment. We deleted the word “predominant”.

4.3. Page 10, the data may support the statement that “depletion of Mcam in the mammary tissue leads to an increase in macrophages”, but it is premature to state “supporting the function of MaSCs”.

Response: Thank you for your suggestions. We deleted “the function of MaSCs”.

4.3 Page 12, “found that systemic ablation of macrophages reversed the increase in MaSC abundance in CKO mice (Figs. 4L-4M),” should read “found that systemic ablation of macrophages reversed the increase in CD24⁺CD29⁺ cell abundance in CKO mice (Figs. 4L-4M),”.

Response: Thank you for your corrections. The statement has been modified

accordingly.

In Figure 2, the authors aim to determine the impact of loss of Mcam on MaSC activity, as measured by mammary transplantation assays. First, they provide a FACS examination of Lin⁻CD24⁺CD29⁺ cells, which revealed 3-fold elevation in Mcam KO mice (about 10% of MECs). The authors identify a difference in MRU frequencies in MECs or Lin⁻CD24⁺CD29^{high} cells between WT and Mcam KO mice. However, both of these frequencies are substantially lower than published frequencies making it difficult to interpret the significance of these findings.

Comment 5 - The authors performed a limiting dilution MRU assay, but identified very low frequencies (Figure 2D, WT : 1/256,161, range: 1:81,640 (mistake) to 1/80,375; CKO : 1/69,591) compared to reported 1:5000 (Shackleton et al., 2006). But, more surprising is the low frequency of MRU observed in WT basal cells (1:1023), which is about 10X lower than reported by Gyorki et al 2009 for Lin⁻CD24⁺CD29^{high} cells (1:97) or Shackleton et al 2006 (1:64). It is hard to determine the cause of this discrepancy as the Methodology states that a similar protocol was used. Please explain, in particular for the transplantation of CD24⁺CD29^{high} basal cells, why the MRU frequencies are only 10% of expected.

Response: Thank you for your comments. For this transplantation assay, the appearance of mammary ducts in the fat pads were considered to calculate success rates. Notably, the earlier the dissection time points of the fat pad, the lower the frequencies of successful duct regeneration. In previous reports, regenerated mammary tissues were harvested during the mature stage (8–12 weeks (Cai et al., 2017) and 6–8 weeks (Chakrabarti et al., 2014)), while we selected the premature stage (5 weeks) for our experiments to capture differences between the Mcam WT and CKO groups. As such, the regeneration frequencies were lower than those reported in the literature. We selected the 5-week-stage as Mcam acts as an inhibitor to delay ductal elongation. At the mature stages, both the WT and CKO groups reached the maximum limits of the fat pad, leading to indistinguishable filled percentages. Earlier reports on mammary duct inhibitors (Bernardo et al., 2010;

Chakrabarti et al., 2014; Shackleton et al., 2006) have also selected the premature stage for assessing endpoints. To confirm this, we repeated the transplantation assay, harvesting the regenerated mammary tissues at a later stage (6 weeks). As shown in Fig. 2F, a higher success rate was achieved, thereby validating our experimental procedures.

Figure 2F Representative images (left) and reconstitution efficiency at limiting dilution (right) of whole mount-stained mammary outgrowths derived from transplantation of WT or CKO basal cells. Scale bar, 1 mm. n = 6 for groups of 2 000, 1 000, and 500 cells. n = 5 for groups of 200 cells.

In Figure 3, the researchers analyze DEGs in Mcam KO versus WT mammary tissues, which indicated potential deregulation of the immune response. Analysis by FACS is not shown but is reported to indicate subtle increases in macrophages (also shown in Figure S4A) and neutrophils. The authors analyse Cx3cr1+ populations by FACS and IF, and these results indicate elevated macrophage levels in Mcam KO tissues although there are concerns with these analyses listed below. But, conditioned medium acquired after culture of CKO MECs augmented the recruitment of macrophages through transwell. Taken together, the data presented in Figure 3 implies a potential recruitment of macrophages in Mcam KO tissues. BUT, the stringency of the protocols and measurements raises questions about the robustness of the subtle differences that are presented.

Comment 6 - It is critical to visualize the FACS profiles and gating strategy used to identify the specified cell types in Figure 3B, even for the cell types that do not show a significant difference between genotypes. Similarly, in Figure 3F, please show the

gating strategy. Why do the *Cx3cr1* profiles appear to measure events of unspecific size scatter?

Response: Thank you for your comments. We have provided all FACS images as suggested, including the gating strategy of all specified cell types in Fig. 3B (Figs. S6B-S6C) and the gating strategy in Fig. 3F (Fig. S7C). The cells were first sorted by size using forward scatter (FSC) in FACS, then the gates were set for the population that contained the target cells, with the proportion of target cells within that specific population finally analyzed.

Figures S6 and S7. (S6B) Gating strategy of dendritic cells (DCs), B cells,

CD4⁺ T cells, and CD8⁺ T cells in FACS. (S6C) Gating strategy of macrophages and neutrophils in FACS. (S7C) Gating strategy of Cx3cr1⁺ macrophages in FACS.

In Fig. 3B, the data correspond to the ratios of total mammary cells. Given the relatively small proportions of immune cell populations in mammary tissues (Fu et al., 2020; Inman et al., 2015), the observed increase was less robust than expected. In our study, the proportion of macrophages in mammary tissue (~5.16% in WT mice) is consistent with previous research demonstrating that a decrease in macrophage proportion from 5.8% to 3.65% corresponded to a decrease in the percentage of Lin⁻CD24⁺CD29⁺ cells from 8.86% to 5.39% (Chakrabarti et al., 2018), suggesting that a mild change in cell number may generate a robust phenotype.

In regard to the Cx3cr1 sorting results, we have showed all FACS images as suggested (Fig. S7C), validating our gating strategy. Based on the published literature (Hasegawa et al., 2019; Koscsó et al., 2020), Cx3cr1 expression levels present as continuous profile, which is consistent with that labelling by single molecular are usually shows as dispersed events, but not isolated populations.

Comment 7 – Similar to Figure 1I and Figure 1K, how were the sections chosen in Figure 3E? The Mcam KO “duct” structure (i.e. multiple cell layers) looks quite distinct from that of the WT duct. That is, the sections do not appear to be at equivalent regions.

Response: Thank you for your comments. In mammary tissues, epithelial structures are easily identifiable, characterized by their encapsulation within adipocyte tissues as well as their closed lumen and compact cell layers. These structures can be labeled by the epithelial cell marker, EpCAM, as shown in previous publications (Martowicz et al., 2013; Sarrío et al., 2012). The image provided for CKO in Fig. 3 represents a partial section of the mammary duct, with the complete image presented in Fig. S7A. We included additional images at various magnifications to demonstrate the similarity in the mammary ducts between the WT and CKO groups (Fig. S7B). Notably, the mammary ducts in the CKO group are thicker than those in the WT

group.

Figure S7. (S7A) Intact picture of original Fig. 3E. (S7B) Immunostaining of Cx3cr1 in WT and CKO mammary tissues. Scale bar, 20 μ m. n = 5 sections in each group.

In Figure 4, the authors examine the importance of macrophages for the observed MaSC activity. Macrophages are well known to affect mammary morphogenesis, repopulating ability, and MaSC activity. Published clonogenic capacity of $CD29^{high}CD24^{+}$ MECs from wildtype glands is about 15 cells per 100 plated using in vitro colony forming assay (Gyorki et al 2009). So, the observed clonogenic capacity of Mcam KO cells using in vitro mammosphere assay (~13) is greater than the observed for wildtype cells (10) but within the range of published wildtype capacities. Moreover, the inhibitory impact of CL treatment on percent fat pad filling is documented (Gyorki et al 2009), and the results presented in Figure 4H are consistent

with this report. However, it is not clearly stated in the Results what tissues were analysed by FACS for Figure 4L-M. The authors indicate CL-treated MECs were implanted in NOD/SCID mice, resulting in no epithelial cells filling the pad (Figure 4K). So, it is not clear what tissue is the source for the CD24+CD29+ epithelial cells shown in the FACS profile in Figure 4L and Figure 4M; could it be CL-treated Mcam KO mice?

Response: Thank you for your comments and questions. Figure 4K and Figs. 4L-4M are from separate assays. Figures 4L-4M are mammary tissues from Mcam WT and CKO mice, while Fig. 4K is from transplanted mice. The gating strategy is shown below (Figs. S8F and 4L). We have also added control data (PBS treatment) in Figs. 4L-4M.

Figure S8 and 4M. Gating strategy of FACS analysis (S8F) and quantification (4M) of basal population in WT and CKO mice treated with or without CL. n = 5 mice in each group.

Comment 8 – Throughout the description of the results, it is important for the authors to state the specifics of the experiments performed either in the results section or in the Figure legends. For instance, it is not clear how the experiment was performed to

give the results presented in Figure 4H. The authors should clearly state the number of CD24+CD29+ cells transplanted into these pads in the Results (I believe it states 200 cells in the methods). Similarly, for Figure 4K, it is not clear how many MECs were injected; judging from Figure 2D it could be 10,000 or 100,000. Finally, for Figure 4L and 4M, it is not clear what is the source of the epithelium as Figure 4K indicates that the treatment with CL prevents fat pad filling.

Response: Thank you for your suggestions and comments. We have added details in the new figure legends.

For the experimental and statistical methods in Figs. 4G-4H, WT and CKO mice were treated with clodronate liposome (CL) and the control groups were treated with PBS (150–170 μ l volume for both CL and PBS) using a tail vein injection every other day for a week before harvesting the mammary gland at 5 weeks of age (body weight ~15–17 g). These experimental details are the same as the assay in Fig. 4D. After harvesting the mammary gland, whole-mount staining was carried out (Fig. 4G). Starting from the very end of the lymph node, the lengths of the mammary duct (red line) and fat pad (black line) were measured, and the proportion of the fat pad filled (length of red line to black line) was calculated, as shown in Fig. 4H (Fig. S8D).

As shown in Fig. 4K, we transplanted 1×10^4 MECs from WT and CKO mice into NOD/SCID mice. The details have been added in the new legends.

Figure S8D. Statistical method for determining the proportion of the mammary duct that fills the fat pad. Red line, length of mammary duct. Black line, length of fat pad.

In Figure 5, the authors demonstrate heightened IL4 expression, which correlates with Stat6 phosphorylation and mediates macrophage migration in vitro and in vivo. Again, however, the experimental details are absent for the analysis of in vivo macrophage infiltration leaving the reader to assume that Figure 5F – 5K are mammary repopulating assays. But, these experiments may be examining intact WT or Mcam KO glands treated with anti-IL4 and, thus, confirm the effect of anti-IL4 on macrophage recruitment (in Figure 5F – 5I). Again, the specific experimental methodology is not clear.

Response: Thank you for these comments and suggestions. In brief, the experiments in Figs. 5F-5I were from intact WT or Mcam KO glands treated with various antibodies, not repopulating assays by transplantation. We have added details in the new figure legends to clarify.

Canonical Wnt signaling pathways did not seem to discriminate between WT and CKO glands but the expression of Wnt5a was dramatically elevated at both the transcript expression and protein expression levels. Next, various cell populations were isolated (but, representative FACS profiles were not provided) and Wnt5a expression was shown to be produced by F4/80+ macrophages isolated in stroma. Macrophages were isolated and treated with a non-hairpin shRNA control or a shRNA targeting Wnt5a and transplanted with CD24+CD29+ cells isolated from either of WT or CKO glands. The efficacy of Wnt5a knockdown is not shown. But, the intervention equivalently blocked the outgrowths from WT or CKO basal cells, indicating an effect that is not specific to the Mcam null gland. Similarly, in Figure 7, the author's present data that indicates the elevated expression of Ryk, and its coexpression with Mcam, in basal cells. Anti-Ryk antibody treatment reduced colony forming capacity for WT and CKO mammary cells, indicating an effect that is not specific to the Mcam null gland.

Response: Thank you for your suggestions and comments.

The FACS profiles for the Lin⁻, luminal, basal, stromal, and CD45⁺F4/80⁺ macrophage cell populations are shown in Fig. S9E. In addition, we also stated the

Wnt5a knockdown efficiency (Fig. S9F).

After knocking down Wnt5a, the enhanced regenerative capacity of the mammary gland induced by Mcam loss was almost entirely inhibited, although a mild inhibitory effect was also observed in WT mice (Fig. 6D), indicating that Wnt5a predominantly works on CKO mice. We agree that Wnt5a also works on WT mice and is required for the other signals, e.g., WNT/Ca²⁺ and WNT/planar cell polarity (PCP) (Zhuang et al., 2017). Although our model established the role of Mcam, other molecules mediated by Wnt5a are also required for mammary gland development.

In addition, when we blocked Ryk using its antibody, the increase in clone number and diameter caused by Mcam loss was suppressed to a level comparable to that in WT mice after Ryk blockade (Figs. 7G-7I), suggesting that the influence of Mcam on MaSCs is Ryk-dependent. We also agree that Ryk antibody treatment reduced clonogenicity in WT mice, indicating that Ryk mediates signals beyond Mcam.

Figure S9. (S9E) Gating strategies of Lin⁻, basal, luminal, stromal, and CD45⁺F4/80⁺ macrophage cell populations in FACS. (S9F) Knockdown efficiency of Wnt5a-shRNAs in basal cells isolated from WT and CKO mice.

- Is the methodology sound? Does the work meet the expected standards in your field?

- Is there enough detail provided in the methods for the work to be reproduced?

In summary, it is very difficult to critically evaluate the experiments, and interpret the

novelty of the findings, because the methodological details are either extremely sparse or completely lacking in the description of the results and in the Figure legends. The presentation of the data generated in the initial sgRNA screen is lacking multiple key controls and the specific process that identified Mcam as a candidate is not clear. As well, the image quality of tissue sections is poor in general and the sections do not seem to be appropriately matched between genotypes. Finally, representative FACS profiles and gating strategies should be included.

Response: Thank you very much. We appreciate your comments and have endeavored to provide comprehensive information and reasoning related to your concerns, which we hope we have been satisfactorily addressed. We thank you again for your critical reading and excellent suggestions, which have helped improve our manuscript.

References

- Azizi, E., Carr, A.J., Plitas, G., Cornish, A.E., Konopacki, C., Prabhakaran, S., Nainys, J., Wu, K., Kiseliovas, V., Setty, M., *et al.* (2018). Single-Cell Map of Diverse Immune Phenotypes in the Breast Tumor Microenvironment. *Cell* *174*, 1293-1308 e36.
- Bernardo, G.M., Lozada, K.L., Miedler, J.D., Harburg, G., Hewitt, S.C., Mosley, J.D., Godwin, A.K., Korach, K.S., Visvader, J.E., Kaestner, K.H., *et al.* (2010). FOXA1 is an essential determinant of ERalpha expression and mammary ductal morphogenesis. *Development* *137*, 2045-2054.
- Cai, S., Kalisky, T., Sahoo, D., Dalerba, P., Feng, W., Lin, Y., Qian, D., Kong, A., Yu, J., Wang, F., *et al.* (2017). A Quiescent Bcl11b High Stem Cell Population Is Required for Maintenance of the Mammary Gland. *Cell Stem Cell* *20*, 247-260. e245.
- Chakrabarti, R., Celia-Terrassa, T., Kumar, S., Hang, X., Wei, Y., Choudhury, A., Hwang, J., Peng, J., Nixon, B., Grady, J.J., *et al.* (2018). Notch ligand Dll1 mediates cross-talk between mammary stem cells and the macrophageal niche. *Science* *360*.
- Chakrabarti, R., Wei, Y., Hwang, J., Hang, X., Andres Blanco, M., Choudhury, A., Tiede, B., Romano, R.A., DeCoste, C., Mercatali, L., *et al.* (2014). DeltaNp63 promotes stem cell activity in mammary gland development and basal-like breast cancer by enhancing Fzd7 expression and Wnt signalling. *Nat Cell Biol* *16*, 1004-1015, 1001-1013.

- Fu, N.Y., Nolan, E., Lindeman, G.J., and Visvader, J.E. (2020). Stem Cells and the Differentiation Hierarchy in Mammary Gland Development. *Physiol Rev* 100, 489-523.
- Hasegawa, T., Kikuta, J., Sudo, T., Matsuura, Y., Matsui, T., Simmons, S., Ebina, K., Hirao, M., Okuzaki, D., Yoshida, Y., *et al.* (2019). Identification of a novel arthritis-associated osteoclast precursor macrophage regulated by FoxM1. *Nat Immunol* 20, 1631-1643.
- Henry, S., Trousdell, M.C., Cyrill, S.L., Zhao, Y., Feigman, M.J., Bouhuis, J.M., Aylard, D.A., Siepel, A., and Dos Santos, C.O. (2021). Characterization of Gene Expression Signatures for the Identification of Cellular Heterogeneity in the Developing Mammary Gland. *J Mammary Gland Biol Neoplasia* 26, 43-66.
- Hovey, R.C., Coder, P.S., Wolf, J.C., Sielken, R.L., Jr., Tisdell, M.O., and Breckenridge, C.B. (2011). Quantitative assessment of mammary gland development in female Long Evans rats following in utero exposure to atrazine. *Toxicol Sci* 119, 380-390.
- Inman, J.L., Robertson, C., Mott, J.D., and Bissell, M.J. (2015). Mammary gland development: cell fate specification, stem cells and the microenvironment. *Development* 142, 1028-1042.
- Kosco, B., Kurapati, S., Rodrigues, R.R., Nedjic, J., Gowda, K., Shin, C., Soni, C., Ashraf, A.Z., Purushothaman, I., Palisoc, M., *et al.* (2020). Gut-resident CX3CR1(hi) macrophages induce tertiary lymphoid structures and IgA response in situ. *Sci Immunol* 5.
- Li, C.M., Shapiro, H., Tsiobikas, C., Selfors, L.M., Chen, H., Rosenbluth, J., Moore, K., Gupta, K.P., Gray, G.K., Oren, Y., *et al.* (2020). Aging-Associated Alterations in Mammary Epithelia and Stroma Revealed by Single-Cell RNA Sequencing. *Cell Rep* 33, 108566.
- Martowicz, A., Rainer, J., Lelong, J., Spizzo, G., Gastl, G., and Untergasser, G. (2013). EpCAM overexpression prolongs proliferative capacity of primary human breast epithelial cells and supports hyperplastic growth. *Mol Cancer* 12, 56.
- O'Brien, J., Lyons, T., Monks, J., Lucia, M.S., Wilson, R.S., Hines, L., Man, Y.G., Borges, V., and Schedin, P. (2010). Alternatively activated macrophages and collagen remodeling characterize the postpartum involuting mammary gland across species. *Am J Pathol* 176, 1241-1255.
- Ruiz, S., Mayor-Ruiz, C., Lafarga, V., Murga, M., Vega-Sendino, M., Ortega, S., and Fernandez-Capetillo, O. (2016). A Genome-wide CRISPR Screen Identifies CDC25A as a Determinant of Sensitivity to ATR Inhibitors. *Mol Cell* 62, 307-313.
- Sarrio, D., Franklin, C.K., Mackay, A., Reis-Filho, J.S., and Isacke, C.M. (2012). Epithelial and mesenchymal subpopulations within normal basal breast cell lines exhibit distinct stem cell/progenitor properties. *Stem Cells* 30, 292-303.
- Shackleton, M., Vaillant, F., Simpson, K.J., Stingl, J., Smyth, G.K., Asselin-Labat, M.L., Wu, L., Lindeman, G.J., and Visvader, J.E. (2006). Generation of a functional mammary gland from a single stem cell. *Nature* 439, 84-88.

- Shalem, O., Sanjana, N.E., Hartenian, E., Shi, X., Scott, D.A., Mikkelsen, T., Heckl, D., Ebert, B.L., Root, D.E., Doench, J.G., *et al.* (2014). Genome-scale CRISPR-Cas9 knockout screening in human cells. *Science* *343*, 84-87.
- Twigger, A.J., Engelbrecht, L.K., Bach, K., Schultz-Pernice, I., Pensa, S., Stenning, J., Petricca, S., Scheel, C.H., and Khaled, W.T. (2022). Transcriptional changes in the mammary gland during lactation revealed by single cell sequencing of cells from human milk. *Nat Commun* *13*, 562.
- Wu, S.Z., Roden, D.L., Wang, C., Holliday, H., Harvey, K., Cazet, A.S., Murphy, K.J., Pereira, B., Al-Eryani, G., Bartonicek, N., *et al.* (2020). Stromal cell diversity associated with immune evasion in human triple-negative breast cancer. *EMBO J* *39*, e104063.
- Zhuang, X., Zhang, H., Li, X., Li, X., Cong, M., Peng, F., Yu, J., Zhang, X., Yang, Q., and Hu, G. (2017). Differential effects on lung and bone metastasis of breast cancer by Wnt signalling inhibitor DKK1. *Nat Cell Biol* *19*, 1274-1285.

Reviewers' comments:

Reviewer #2 (Remarks to the Author):

The authors have made substantial revisions to address many of the comments regarding their initial submission. These revisions include: i. the inclusion of additional control data to allow the reader to better interpret the results of their library screen; ii. new data describing the expression of *Mcam* in the young, older, pregnant, lactating, and involuting mouse mammary glands; iii. repetition of transplantation experiments with a later timepoint for analysis, which resulted in higher success rates and inferred MRU frequencies that are consistent with the published literature; iv. the authors created and characterized K14-Cre *Mcam* flox/wt mice; and, v. the authors include FACS profiles for the measurement of immune cell infiltrations.

Although I feel that the revised manuscript is improved, the authors must clarify some critical details. It is essential that the authors provide a clear description of the genotype for their CKO mice. Because the genotype was not explicitly stated, I assume CKO mice are MMTV-Cre *Mcam* flox/wt.

- Should these CKO mice be heterozygous for the *Mcam* flox allele, similar to the K14-Cre *Mcam* flox/wt mice, the authors must justify their study of heterozygous *Mcam* conditional knockout mice. Given the mammary gland develops postnatally, homozygous *Mcam* conditional knockout mice should be viable and the mammary gland phenotype of these animals should be described.
- Should these CKO mice be MMTV-Cre *Mcam* flox/flox mice, the authors should discuss the interesting result that heterozygous loss of *Mcam* (in K14-Cre *Mcam* f/wt mice) phenocopies homozygous loss of *Mcam* in CKO mice.

Major Comment 1 –

The authors must clearly state the genotype of the MMTV-Cre *Mcam* flox knockout mice. The authors only state (line 141) “Then we bred *Mcam* flox mice (Zeng et al., 2014) with MMTV-Cre mice (Wagner et al., 1997) ...” and (Methods, line 433) “The *Mcam* floxed mice have been described previously (Zeng et al., 2014).” Therefore, it is not clear whether the authors are using MMTV-Cre *Mcam* flox/flox mice or MMTV-Cre *Mcam* flox/wt mice.

The authors refer to MMTV-Cre *Mcam* flox mice by the acronym “CKO”. As the authors also use CKO on line 92 to mean CRISPR-Cas9 Knockout, the authors should change the acronym from “CKO” to “cKO”, and define the mice as conditional knockout (cKO). At least for the K14-Cre *Mcam* flox/wt mice, there is one wildtype allele so these mice should be defined as heterozygous *Mcam* cKO mice.

Major Comment 1b –

Because the genotype is not stated explicitly, I assume that the genotype for the MMTV-Cre *Mcam* knockout mice (so-called CKO mice) is MMTV-Cre *Mcam* flox/wt. I make this assumption because many of their phenotypes are conserved in the K14-Cre *Mcam* flox/wt mice, which were generated for the revision experiments.

If CKO mice are similar to K14-Cre *Mcam* flox/wt mice, in that these mice express a wildtype allele, I believe it is important for the authors to describe the mammary gland phenotype for K14-Cre and/or MMTV-Cre *Mcam* flox/flox mice (homozygous cKO mice). Does complete homozygous knockout of *Mcam* (*Mcam* flox/flox) produce a different phenotype from heterozygous knockdown of *Mcam* (*Mcam* flox/wt)?

If CKO mice are MMTV-Cre *Mcam* flox/flox mice, please explain why K14-Cre *Mcam* flox/wt mice share many similar phenotypes with MMTV-Cre *Mcam* flox/flox mice.

Other comments for the authors to address.

Comment 1 - For the Genome-scale CRISPR-Cas9 KnockOut (Ge-CKO) library screen, the authors should clarify the criteria leading to their selection of *Mcam* from the hundreds or thousands of potential sgRNA that were enriched. According to Figure S2A, *Mcam* was one of 31 gene products that were prioritized based upon “enrichment in basal lineage”. As the selection of *Mcam* appears to be largely based upon the enrichment of its expression in basal lineage (as shown in Figure S2A), it seems redundant to then characterize *Mcam* expression in the basal lineage, as shown in Figure 1B, 1C, 1D.

Comment 2 – I appreciate the authors inclusion of additional control data for the Ge-CKO screen. But, I disagree with their interpretation on line 113 that Fig. S1B indicates that “sgRNA were highly

enriched in (transplantation) samples". With the exception of mouse S5979, sgRNA enrichment was modest or minimal for large clones, mouse S5977, mouse S5975, mouse S5974, and mouse S5973 with respect to cells (appreciating the log scale for the graph). Please remove the word "highly" from line 113.

Comment 3 – Many of the displayed immunofluorescence images in the revised manuscript are of low resolution and should be improved. This may be due to the pdf quality but there are other issues, as well. For instance, in Figure 1E, the images have inconsistent magnifications – making comparisons between images difficult – and the DAPI intensities are also inconsistent. In Figure 2A, the resolution of the images is poor so it is difficult to evaluate the staining patterns.

Comment 4 – line 182 – In the results section, it is important to state the differences in the experimental procedures for the results presented in Figure 2E and Supplemental Figure 5D, which differ due to the time of harvest (week 5 or week 6, respectively).

Comment 5 – Figure 2 demonstrates that the loss of Mcam results in a significant increase (3-fold) in basal cells. When those basal cells are transplanted, they observe a significant increase in outgrowths. Moreover, when those basal cells are subjected to an in vitro clone assay, the cells show an increase in the capacity to generate colonies and those individual colonies are larger. These latter experiments suggest a basal cell-intrinsic mechanism for increased clonogenicity following the loss of Mcam expression. A basal cell-intrinsic mechanism appears to be independent of the macrophage-mediated mechanism that is described in the manuscript. Minor point – how were the clones defined (and subsequently counted) as there seems to be quite heterogeneous sizes in the images (Figure 2G).

Comment 6 – line 211 and Fig S6B – I appreciate the inclusion of the FACS profiles. But, the authors must justify the alteration in the SSC-FSC gate for their analysis of T cells. The very small gate is quite different from those used to measure B cells, DC, macrophages, and neutrophils.

Comment 7 – line 217 and Fig 3F, 3G and Fig S7A, S7B – I do not find the measurement of Cx3cr1 in CKO vs WT (FACS plots) to be convincing, but I do appreciate the additional control data provided by the authors. Still, there does not seem to be a population gated. Similarly, I do not find the measurement of Cx3cr1 in CKO vs WT (IF) or in Mcam flox/wt vs Mcam flox/wt K14-Cre (IF) to be convincing, but I do appreciate the additional images provided by the authors. The IF staining for Cx3cr1 seems to be non-specifically enriched in the epithelia.

Comment 8 – line 231 and Fig 4A, 4C – The clone diameter measurements in Fig. 4C indicate the clones in Fig. 4A (90 – 140 μm) are smaller than the clones presented in Fig 2G (100 – 300 μm). The figure legends indicate the scale bars are 100 μm in both Figure 2G and Figure 4A. But, the images in Figure 4A show much LARGER clones than shown in Figure 2G. How is this possible? In fact, none of the CKO clones shown in Figure 2A appear to be larger than the 100 μm scale bar. Please explain why the CKO clones in Figure 2A do not appear to be 100 to 300 μm in diameter.

Comment 9 – line 299 – The level of expression of Wnt5a in control tissues (Mcam flox/wt) appears to be highly variable. In the image presented in Figure 6B (WT panel), there is little or no staining. In the image presented in Figure S9D (Mcam flox/wt panel), however, there is moderate staining in luminal epithelia. The authors should explain this discrepancy.

Comment 10 – line 335, Figure 7C – the mammary tissue sections used to show co-localization of Ryk and Mcam or Ryk and SMA are poorly selected. One of the sections does not look ductal or alveolar (LHS) and the other section looks to have a filled lumen (RHS).

Reviewers' comments:

Reviewer #2 (Remarks to the Author):

The authors have made substantial revisions to address many of the comments regarding their initial submission. These revisions include: i. the inclusion of additional control data to allow the reader to better interpret the results of their library screen; ii. new data describing the expression of Mcam in the young, older, pregnant, lactating, and involuting mouse mammary glands; iii. repetition of transplantation experiments with a later timepoint for analysis, which resulted in higher success rates and inferred MRU frequencies that are consistent with the published literature; iv. the authors created and characterized K14-Cre Mcam flox/wt mice; and, v. the authors include FACS profiles for the measurement of immune cell infiltrations.

Response: We thank Reviewer 2 for positive summary of our major revisions.

Although I feel that the revised manuscript is improved, the authors must clarify some critical details. It is essential that the authors provide a clear description of the genotype for their CKO mice. Because the genotype was not explicitly stated, I assume CKO mice are MMTV-Cre Mcam flox/wt.

• Should these CKO mice be heterozygous for the Mcam flox allele, similar to the K14-Cre Mcam flox/wt mice, the authors must justify their study of heterozygous Mcam conditional knockout mice. Given the mammary gland develops postnatally, homozygous Mcam conditional knockout mice should be viable and the mammary gland phenotype of these animals should be described.

• Should these CKO mice be MMTV-Cre Mcam flox/flox mice, the authors should discuss the interesting result that heterozygous loss of Mcam (in K14-Cre Mcam f/wt mice) phenocopies homozygous loss of Mcam in CKO mice.

Response: Thank you for your recognition of our revision and new suggestions.

In this study, the CKO (conditional knockout) was applied for abbreviation of

homozygous knockout by MMTV-Cre (Mcam^{fl/fl}/MMTV-Cre). For the depletion by K14-Cre, it is heterozygous knockout (Mcam^{fl/wt}/K14-Cre). We have added the description in the revised manuscript. We apologize for the missing clear description of the genotype for CKO mice.

Since MMTV promoter is specifically activated in mammary gland (Wagner et al., 1997), which is well-known in mammary gland study, the homozygous depletion of Mcam did not induce lethal or any other phenotypes outside of mammary gland.

Both Cre lines driven by MMTV and K14 promoters are designed for the depletion of targeted genes in mammary gland. The MMTV promoter is chimerically activated in mammary epithelial cells (Wagner et al., 1997), while K14 promoter is specifically activated in basal cells and much more robust (Jonkers et al., 2001). Moreover, Mcam is predominantly expressed in the basal layer of the mammary gland, the phenotypes using K14-cre mice were therefore much more intense than those by MMTV-Cre mice.

As to why heterozygous loss of Mcam by K14-Cre (Mcam^{fl/wt}/K14-Cre) had similar phenotypes to homozygous loss by MMTV-Cre mice, it is due to the similar protein levels of Mcam (approximately 30% to their controls) (Figure 1H and Figure S3E) between two genotypes.

Percentages of Mcam protein levels to α -tubulin (left) and their raw gray intensities (right) of Figures 1H and S3E.

Major Comment 1 –

The authors must clearly state the genotype of the MMTV-Cre Mcam flox knockout

mice. The authors only state (line 141) “Then we bred Mcam flox mice (Zeng et al., 2014) with MMTV-Cre mice (Wagner et al., 1997) ...” and (Methods, line 433) “The Mcam floxed mice have been described previously (Zeng et al., 2014).” Therefore, it is not clear whether the authors are using MMTV-Cre Mcam flox/flox mice or MMTV-Cre Mcam flox/wt mice.

The authors refer to MMTV-Cre Mcam flox mice by the acronym “CKO”. As the authors also use CKO on line 92 to mean CRISPR-Cas9 Knockout, the authors should change the acronym from “CKO” to “cKO”, and define the mice as conditional knockout (cKO). At least for the K14-Cre Mcam flox/wt mice, there is one wildtype allele so these mice should be defined as heterozygous Mcam cKO mice.

Response: Thank you for your questions and suggestions.

As the responses in last question, the “CKO” is homozygous depletion of Mcam (Mcam^{fl/fl}/MMTV-Cre). The specific descriptions about genotypes of the CKO mice have been added to lines 142-144.

According to the reviewer’s suggestion, we have changed the acronym from “CKO” to “cKO”. For Mcam^{fl/wt}/K14-Cre mice, we redefined it as heterozygous cKO mice.

Major Comment 1b –

Because the genotype is not stated explicitly, I assume that the genotype for the MMTV-Cre Mcam knockout mice (so-called CKO mice) is MMTV-Cre Mcam flox/wt. I make this assumption because many of their phenotypes are conserved in the K14-Cre Mcam flox/wt mice, which were generated for the revision experiments.

If CKO mice are similar to K14-Cre Mcam flox/wt mice, in that these mice express a wildtype allele, I believe it is important for the authors to describe the mammary gland phenotype for K14-Cre and/or MMTV-Cre Mcam flox/flox mice (homozygous cKO mice). Does complete homozygous knockout of Mcam (Mcam flox/flox) produce a different phenotype from heterozygous knockdown of Mcam (Mcam flox/wt)?

If CKO mice are MMTV-Cre Mcam flox/flox mice, please explain why K14-Cre Mcam

flox/wt mice share many similar phenotypes with MMTV-Cre Mcam flox/flox mice.

Other comments for the authors to address.

Response: Thank you for your suggestions.

As the responses in last question, the “CKO” is homozygous depletion of Mcam (Mcam^{fl/fl}/MMTV-Cre).

For the discussion why heterozygous loss of Mcam by K14-Cre (Mcam^{fl/wt}/K14-Cre) had similar phenotypes to homozygous loss by MMTV-Cre mice, please see the detailed responses for the first question.

Comment 1 - For the Genome-scale CRISPR-Cas9 KnockOut (Ge-CKO) library screen, the authors should clarify the criteria leading to their selection of Mcam from the hundreds or thousands of potential sgRNA that were enriched. According to Figure S2A, Mcam was one of 31 gene products that were prioritized based upon “enrichment in basal lineage”. As the selection of Mcam appears to be largely based upon the enrichment of its expression in basal lineage (as shown in Figure S2A), it seems redundant to then characterize Mcam expression in the basal lineage, as shown in Figure 1B, 1C, 1D.

Response: Thank you for your comments.

As we mentioned in Page 6, after bioinformatics analysis, the criteria is to choose the common genes with mammary gland morphogenesis and/or MEC clonogenicity and regenerative capacity.

The expression profile is crucial and fundamental for supporting the function of Mcam, it is therefore very necessary and important to verify the reliability of the sequencing results. We carried out the experimental examination of Mcam expression in the basal lineage, and Figure 1B-1D are indispensable for enhancing the reliability of the sequencing results.

Comment 2 – I appreciate the authors inclusion of additional control data for the Ge-CKO screen. But, I disagree with their interpretation on line 113 that Fig. S1B

indicates that “sgRNA were highly enriched in (transplantation) samples”. With the exception of mouse S5979, sgRNA enrichment was modest or minimal for large clones, mouse S5977, mouse S5975, mouse S5974, and mouse S5973 with respect to cells (appreciating the log scale for the graph). Please remove the word “highly” from line 113.

Response: Thank you for your suggestion. We have removed the word “highly” as required.

Comment 3 – Many of the displayed immunofluorescence images in the revised manuscript are of low resolution and should be improved. This may be due to the pdf quality but there are other issues, as well. For instance, in Figure 1E, the images have inconsistent magnifications – making comparisons between images difficult – and the DAPI intensities are also inconsistent. In Figure 2A, the resolution of the images is poor so it is difficult to evaluate the staining patterns.

Response: Thank you for your suggestions.

We have demonstrated the images with the same magnification in Figure 1E. The images with higher resolution in Figure 2A have been demonstrated.

Comment 4 – line 182 – In the results section, it is important to state the differences in the experimental procedures for the results presented in Figure 2E and Supplemental Figure 5D, which differ due to the time of harvest (week 5 or week 6, respectively).

Response: We have added the time of harvest in Figure 2E and Supplemental Figure 5D in both the result section and the figure legend.

Comment 5 – Figure 2 demonstrates that the loss of Mcam results in a significant increase (3-fold) in basal cells. When those basal cells are transplanted, they observe a significant increase in outgrowths. Moreover, when those basal cells are subjected to an in vitro clone assay, the cells show an increase in the capacity to generate colonies and those individual colonies are larger. These latter experiments suggest a

basal cell-intrinsic mechanism for increased clonogenicity following the loss of Mcam expression. A basal cell-intrinsic mechanism appears to be independent of the macrophage-mediated mechanism that is described in the manuscript. Minor point – how were the clones defined (and subsequently counted) as there seems to be quite heterogeneous sizes in the images (Figure 2G).

Response: Thank you for your comments and suggestions.

Besides dominant source of macrophage, Wnt5a has a small amount of intracellular expression in epithelial cells (Figure 6C), providing the potential basal cell-intrinsic mechanism to sustain the clonogenicity.

For the clone formation assay, all the colonies were counted using Image J software, which is based on the textbook of methods and protocols of mammary gland development (Shehata and Stingl, 2017). The method was widely applied for mammary gland studies (Liu et al., 2017; Zeng et al., 2016).

Comment 6 – line 211 and Fig S6B – I appreciate the inclusion of the FACS profiles. But, the authors must justify the alteration in the SSC-FSC gate for their analysis of T cells. The very small gate is quite different from those used to measure B cells, DC, macrophages, and neutrophils.

Response: Thank you for your recognition of the inclusion of the FACS profiles and raised new questions.

The gating strategy about the SSC-FSC for analysis of various immune cells is carried out followed the Flow Cytometry Guidelines (Cossarizza et al., 2021), which define T cell populations with a small gate, and the other immune cells, including B cells, macrophages, and dendritic cells, with larger gates. These gating strategies were also used in the published literature widely (He et al., 2023; Zhou et al., 2020).

[redacted]

Gating strategy of CD4 and CD8 T cells from the spleen (Page 50, Flow Cytometry Guidelines (Cossarizza et al., 2021)).

[redacted]

Gating strategy for B cells. (Page 223, Flow Cytometry Guidelines (Cossarizza et al., 2021)).

[redacted]

Gating strategy for the macrophages. (Page 312, Flow Cytometry Guidelines (Cossarizza et al., 2021)).

[redacted]

Gating strategy for dendritic cells. (Page 316, Flow Cytometry Guidelines (Cossarizza et al., 2021)).

Comment 7 – line 217 and Fig 3F, 3G and Fig S7A, S7B – I do not find the measurement of Cx3cr1 in CKO vs WT (FACS plots) to be convincing, but I do appreciate the additional control data provided by the authors. Still, there does not seem to be a population gated. Similarly, I do not find the measurement of Cx3cr1 in CKO vs WT (IF) or in Mcam flox/wt vs Mcam flox/wt K14-Cre (IF) to be convincing, but I do appreciate the additional images provided by the authors. The IF staining for Cx3cr1 seems to be non-specifically enriched in the epithelia.

Response: Thank you for your suggestions.

Figure 3F is labelled by single antibody, which is always indicated as dispersed events, but not isolated populations. In the published literatures (Ahadzadeh et al., 2018; Koscsó et al., 2020), as shown in the images below, Cx3cr1 expression levels

are also presented as continuous profile, but not as population gated. Our results showed the similar pattern (Figure S7C). With the same gating conditions, the results are convincing for the remarkable elevation of population ratios.

[redacted]

Flow cytometry plots define phenotypes of all mononuclear phagocytes (MPs) subsets in the normal (day 0) and infected (day 10 after infection) colon (Koscso et al., 2020).

Cx3cr1 is a marker of tissue-resident macrophage marker (Dawson et al., 2020), and in this published research, the authors stated that tissue-resident macrophages (Cx3cr1⁺ cells) located at inner myoepithelial surface and intra-epithelial localization. In addition, Cx3cr1 has also been reported to be expressed in dendritic cells, macrophages and other cells.

Comment 8 – line 231 and Fig 4A, 4C – The clone diameter measurements in Fig. 4C indicate the clones in Fig. 4A (90 – 140 μm) are smaller than the clones presented in Fig 2G (100 – 300 μm). The figure legends indicate the scale bars are 100 μm in both Figure 2G and Figure 4A. But, the images in Figure 4A show much LARGER clones than shown in Figure 2G. How is this possible? In fact, none of the CKO clones shown in Figure 2A appear to be larger than the 100 μm scale bar. Please explain why the CKO clones in Figure 2A do not appear to be 100 to 300 μm in diameter.

Response: Thank you for your questions.

We re-checked all the pictures for Figure 2G (as shown below), and found the scale bar should be 500 μm instead of 100 μm , and Figure 2I demonstrates the accurate statistical results. We have corrected the mistake in the corresponding figure

legend, and appreciate Reviewer 2 for the critical readings.

2G

Figure 2G. Representative images of colonies formed by MECs derived from WT and cKO mice. Scale bar, 500 μ m.

Comment 9 – line 299 – The level of expression of Wnt5a in control tissues (Mcam flox/wt) appears to be highly variable. In the image presented in Figure 6B (WT panel), there is little or no staining. In the image presented in Figure S9D (Mcam flox/wt panel), however, there is moderate staining in luminal epithelia. The authors should explain this discrepancy.

Response: Thank you for your comments.

The staining color depends on the incubation time with its substrate DAB (Diaminobenzidine). Longer the incubation time, darker the brown color developed. Considered the mammary ducts in WT have basal expression levels of Wnt5a (Figure 6C), the different batches of staining assays generate variation of staining color. Moreover, Wnt5a have variation of expression among different mouse individuals, which is quite common in mouse genetic experiments.

The overall objective here is to verify the elevation of Wnt5a expression in Mcam knockout mice. Our experiments always demonstrate the consistent elevation of Wnt5a in the mammary gland of cKO mice (as evidenced by multiple pairs of staining results below), strongly supporting our conclusion.

Figure 6B. IHC staining of Wnt5a in WT and cKO mice. Scale bar, 20 μ m.

Figure S9D. IHC staining of Wnt5a in Mcam^{fl/wt} and Mcam^{fl/wt}/K14-Cre mice. Scale bar, 50 μ m.

Comment 10 – line 335, Figure 7C – the mammary tissue sections used to show co-localization of Ryk and Mcam or Ryk and SMA are poorly selected. One of the sections does not look ductal or alveolar (LHS) and the other section looks to have a filled lumen (RHS).

Response: Thank you for your suggestion. We have provided the typical mammary duct sections.

Figure 7C. IF images of co-staining of indicated antibodies in WT mammary tissues. Scale bar, 5 μ m.

References

- Ahadzadeh, E., Rosendahl, A., Czesla, D., Steffens, P., Prussner, L., Meyer-Schwesinger, C., Wanner, N., Paust, H.J., Huber, T.B., Stahl, R.A.K., *et al.* (2018). The chemokine receptor CX(3)CR1 reduces renal injury in mice with angiotensin II-induced hypertension. *Am J Physiol Renal Physiol* 315, F1526-F1535.
- Cossarizza, A., Chang, H.D., Radbruch, A., Abrignani, S., Addo, R., Akdis, M., Andra, I., Andreatta, F., Annunziato, F., Arranz, E., *et al.* (2021). Guidelines for the use of flow cytometry and cell sorting in immunological studies (third edition). *Eur J Immunol* 51, 2708-3145.
- He, M., Roussak, K., Ma, F., Borchering, N., Garin, V., White, M., Schutt, C., Jensen, T.I., Zhao, Y., Iberg, C.A., *et al.* (2023). CD5 expression by dendritic cells directs T cell immunity and sustains immunotherapy responses. *Science* 379, eabg2752.
- Jonkers, J., Meuwissen, R., van der Gulden, H., Peterse, H., van der Valk, M., and Berns, A. (2001). Synergistic tumor suppressor activity of BRCA2 and p53 in a conditional mouse model for breast cancer. *Nat Genet* 29, 418-425.
- Kosco, B., Kurapati, S., Rodrigues, R.R., Nedjic, J., Gowda, K., Shin, C., Soni, C., Ashraf, A.Z., Purushothaman, I., Palisoc, M., *et al.* (2020). Gut-resident CX3CR1(hi) macrophages induce tertiary lymphoid structures and IgA response in situ. *Sci Immunol* 5.
- Liu, W., Wu, T., Dong, X., and Zeng, Y.A. (2017). Neuropilin-1 is upregulated by Wnt/beta-catenin signaling and is important for mammary stem cells. *Sci Rep* 7, 10941.
- Shehata, M., and Stingl, J. (2017). Purification of Distinct Subsets of Epithelial Cells from Normal Human Breast Tissue. *Methods Mol Biol* 1501, 261-276.
- Wagner, K.U., Wall, R.J., St-Onge, L., Gruss, P., Wynshaw-Boris, A., Garrett, L., Li, M., Furth, P.A., and Hennighausen, L. (1997). Cre-mediated gene deletion in the mammary gland. *Nucleic Acids Res* 25, 4323-4330.
- Zeng, L., Cai, C., Li, S., Wang, W., Li, Y., Chen, J., Zhu, X., and Zeng, Y.A. (2016). Essential Roles of Cyclin Y-Like 1 and Cyclin Y in Dividing Wnt-Responsive Mammary Stem/Progenitor Cells. *PLoS Genet* 12, e1006055.
- Zhou, H., Zhang, Z., Liu, G., Jiang, M., Wang, J., Liu, Y., and Tai, G. (2020). The Effect of Different Immunization Cycles of a Recombinant Mucin1-Maltose-Binding Protein Vaccine on T Cell Responses to B16-MUC1 Melanoma in Mice. *Int J Mol Sci* 21.

REVIEWERS' COMMENTS

Reviewer #2 (Remarks to the Author):

The authors have addressed my prior comments.

I suggest that the authors include in the article the data comparison presented in this revision/rebuttal of similar Mcam expression in tissues isolated from K14-Cre Mcam fl/wt mice and tissues isolated from MMTV-Cre Mcam fl/fl mice. Moreover, the authors should discuss the interesting result that heterozygous loss of Mcam in K14-Cre Mcam fl/wt mice phenocopies homozygous loss of Mcam in MMTV-Cre Mcam fl/fl (cKO) mice. In their rebuttal, the authors present an argument that the total protein levels in the tissues are similar and thus the phenotypes are similar. But, the total Mcam protein level in the MMTV-Cre Mcam fl/fl animals would be expected to be produced by the non-epithelial stroma, which lack MMTV-Cre mediated recombination. In contrast, the total Mcam protein in the K14-Cre Mcam fl/wt tissues would be expected to be produced by the non-epithelial stroma, the K14-negative (luminal) epithelia, and the K14-positive epithelia albeit at a reduced level. It is hard to envision similar Mcam protein levels are expressed in the basal epithelial cells in these two genotypes. Indeed, the gene expression measurements in MECs do not appear to be equivalent (Fig. 1F and S3c). I believe this is an important issue that should be discussed.

Reviewer #2 (Remarks to the Author):

The authors have addressed my prior comments.

I suggest that the authors include in the article the data comparison presented in this revision/rebuttal of similar Mcam expression in tissues isolated from K14-Cre Mcam fl/wt mice and tissues isolated from MMTV-Cre Mcam fl/fl mice.

Response: We thank the recognition of our revision and new suggestions.

We have included the data comparison of similar Mcam expression in Figure S10e-f based on the reviewer's suggestion.

Moreover, the authors should discuss the interesting result that heterozygous loss of Mcam in K14-Cre Mcam fl/wt mice phenocopies homozygous loss of Mcam in MMTV-Cre Mcam fl/fl (cKO) mice. In their rebuttal, the authors present an argument that the total protein levels in the tissues are similar and thus the phenotypes are similar.

Response: We thank this suggestion and have discussed the results mentioned in the 2nd paragraph of Page 19 of the revised text.

But, the total Mcam protein level in the MMTV-Cre Mcam fl/fl animals would be expected to be produced by the non-epithelial stroma, which lack MMTV-Cre mediated recombination. In contrast, the total Mcam protein in the K14-Cre Mcam fl/wt tissues would be expected to be produced by the non-epithelial stroma, the K14-negative (luminal) epithelia, and the K14-positive epithelia albeit at a reduced level. It is hard to envision similar Mcam protein levels are expressed in the basal epithelial cells in these two genotypes. Indeed, the gene expression measurements in MECs do not appear to be equivalent (Fig. 1F and S3c). I believe this is an important issue that should be discussed.

Response: We thank the critical reading. As this reviewer's description, the previous Mcam expression comparison of Mcam was from total breast tissues or total epithelia (MECs). Actually, in particular basal layer, we also detected the expression

profile of these two genotypes, IF staining showed the similar Mcam depletion between heterozygous and homozygous knockout of Mcam (Figs. 1g and S3d). As to why heterozygous loss of Mcam has the similar expression profile to homozygous loss, this could be due to the expression bias¹ of floxed allele of Mcam. In previous reports, heterozygous loss of Brca1 or Tet2 had less than 10% expression levels of their control groups, which were very close to the homozygous loss of these genes^{2,3}, demonstrating the expression bias of these floxed alleles.

As suggested by this reviewer, we have discussed this issue in the Discussion (Pages 19-20).

References

- 1 Kravitz, S. N. *et al.* Random allelic expression in the adult human body. *Cell Rep* **42**, 111945 (2023).
- 2 Bai, F. *et al.* BRCA1 suppresses epithelial-to-mesenchymal transition and stem cell dedifferentiation during mammary and tumor development. *Cancer Res* **74**, 6161-6172 (2014).
- 3 Kim, M. R., Wu, M. J., Zhang, Y., Yang, J. Y. & Chang, C. J. TET2 directs mammary luminal cell differentiation and endocrine response. *Nat Commun* **11**, 4642 (2020).